# From Specificity to Generality: Revisiting Generalizable Artifacts in Detecting Face Deepfakes

**Long Ma**[1]    **Zhiyuan Yan**[2]    **Jin Xu**[4]    **Yize Chen**[3]
**Qinglang Guo**[1]    **Zhen Bi**[5 6]    **Yong Liao**[1 *]    **Hui Lin**[7]

[1]School of Cyber Science and Technology, University of Science and Technology of China
[2]School of Electronic and Computer Engineering, Peking University
[3]School of Data Science, The Chinese University of Hong Kong
[4]School of Information Science and Technology, University of Science and Technology of China
[5]Huzhou University [6] Banbu AI Foundation
[7]China Academy of Electronics and Information Technology
{longm@mail, yliao@}ustc.edu.cn

## Abstract

Detecting deepfakes has been an increasingly important topic, especially given the rapid development of AI generation techniques. In this paper, we ask: How can we build a universal detection framework that is effective for most facial deepfakes? One significant challenge is the wide diversity of existing deepfake generators, which produced varied types of forgery artifacts (*e.g.,* lighting inconsistency, color mismatch, *etc*). But should we "teach" the detector to learn all these artifacts separately? It is impossible and impractical to elaborate on them all. So the core idea is to pinpoint the more common and general artifacts across different deepfakes. Through systematic analysis of shared technical frameworks in existing deepfake algorithms, we categorize deepfake artifacts into two distinct yet complementary types: Face Inconsistency Artifacts (FIA) and Up-Sampling Artifacts (USA). FIA arise from the challenge of generating all intricate details, inevitably causing inconsistencies between the complex facial features and relatively uniform surrounding areas. USA, on the other hand, are the inevitable traces left by the generator's decoder during the up-sampling process, with all existing deepfakes exhibiting either or both artifacts. Subsequently, we propose a novel image-level pseudo-fake creation framework that constructs fake samples with only the FIA and USA, without introducing extra less-general artifacts. Specifically, we reconstruct the target face to simulate the USA, while utilize image-level blend on diverse facial regions to create the FIA. We surprisingly found that, with this intuitive design, a standard image classifier trained only with our pseudo-fake data can non-trivially generalize well to unseen deepfakes.

## 1 Introduction

In recent years, A growing number of facial manipulation techniques have advanced due to the rapid development of generative models [35, 36, 58, 75], which facilitate the production of highly realistic and virtually undetectable alterations. As a result, the risks of personal privacy being violated and the spread of misinformation have increased significantly. Therefore, developing a universal deepfake detector that can be used to detect the existing diverse fake methods has become an urgent priority.

---

*Corresponding Author.

39th Conference on Neural Information Processing Systems (NeurIPS 2025).

**(a). Face Inconsistency Artifacts (FIA)**

**Face-level Inconsistencies** | **Region-level Inconsistencies**

Real | Fake | Real | DDPM | Eyes | Teeth | Eyebrow

More Details | Less Details | More Details | Less Details | Mouth | Nose | Forehead

**Hint-1:** AI-generated faces often exhibit inconsistencies at both the *face level* and *region level* compared to the surrounding areas.

**(b). Up-Sampling Artifacts (USA)**

Real | DeepFake | Super-Resolution

**Hint-2:** AI-generated faces often exhibit *up-sampling artifacts* by the decoders during the generation process.

Figure 1: Illustration of the two identified general forgery artifacts across different face deepfakes: (a) Face Inconsistency Artifacts and (b) Up-Sampling Artifacts. We show that the existing face deepfakes typically exhibit both or one of these two artifacts.

In this paper, we pose the question: *How can we build a universal detection framework that is effective for most facial deepfakes?* A significant challenge lies in the wide range of deepfake generators, which lead to different types of forgery artifacts. Specifically, a large number of existing works are dedicated to creating detection algorithms that are carefully designed to identify counterfeit artifacts within specific handcrafted designs, including eye blinking [42], pupil morphology [25], and corneal specularity [32]. However, these methods are mainly effective for identifying particular forgery techniques. But should we train the detector to learn each of these artifacts separately? It is impossible and impractical to cover them all. So, *where should we start from?*

The core idea is to encourage the detection model to learn the more generalizable artifacts. As indicated in previous works [81, 63, 70], different fakes might share some common forgery patterns, and the unlimited deepfake artifacts could be generalized and summarized by the limited generalizable artifacts. Motivated by this, we conduct an in-depth preliminary exploration of the common patterns of existing deepfake artifacts, through which we identify two generalizable deepfake artifacts and categorize them into two distinct yet complementary categories: Face Inconsistency Artifacts (FIA) and Up-Sampling Artifacts (USA), as depicted in Figure 1. FIA represents an inherent limitation that deepfakes struggle to overcome, originating from the generator's incapacity to precisely reproduce facial attributes and nuances, as well as the inescapable disparities between regions induced by amalgamation processes. Unlike FIA, Up-Sampling Artifacts (USA) are not readily apparent to the unaided eye and originate from the generator's inherent up-sampling process that the generator's decoder cannot adequately substitute. Crucially, since decoding and up-sampling constitute fundamental operations in facial manipulation generators, such artifacts become an inevitable byproduct across all methods in this domain. Numerous prior investigations have substantiated this observation [22, 65]. To date, all existing deepfakes characteristically display one or both of these artifact categories.

To enable the model to simultaneously capture FIA and USA, we propose a sophisticated data augmentation method: **FIA-USA**, which leverages super-resolution models [71] and autoencoders [57] to introduce Up-Sampling Artifacts (USA) by reconstructing the face, and employs a strategy of generating multiple masks to blend the reconstructed face with the original, thereby introducing Face Inconsistency Artifacts (FIA) by creating discrepancies at both the facial and regional levels. Augmented with the aforementioned dual artifacts, our method effectively conditions the model to generalize across unseen deepfakes. Furthermore, to maximize the efficacy of FIA-USA, we introduce two complementary technique: **Automatic Forgery-aware Feature Selection (AFFS)**, which streamlines feature dimensionality and augments feature discernment through adaptive selection. **Region-aware Contrastive Regularization (RCR)**, by juxtaposing the features of the manipulated

region against those of the authentic region, RCR enhances the model's focus on the upsampling traces within the manipulated areas and the inconsistencies between different regions.

Extensive experiments conducted on seven widely used Deepfake detection datasets (**encompassing over 58 distinct forgery methods, spanning facial manipulation categories including identity swapping, expression reenactment, generative face synthesis, and attribute editing**) validate the superior efficacy of our method. Detailed ablation experiments and robustness experiments have demonstrated the effectiveness of each component in our method and the robustness of our method to disturbances.

Table 1: **Comparison of our framework and state-of-the-art(SOTA) methods using pseudo-deepfake synthesis.** Our method differs from previous methods in terms of the level at which alterations are applied (local vs global), the introduced artifacts, and the level at which facial inconsistencies are applied.

| Methods | Alteration artifact | | Introduced Artifacts | | |
|---|---|---|---|---|---|
| | global | local | Face Inconsistency | | Up-Sampling |
| | | | face-level | region-level | |
| Face Xray [40], PCL [89], OST [4] | ✓ | | ✓ | | |
| SLADD [3] | ✓ | | | ✓ | |
| SBI [63] | ✓ | | ✓ | | |
| SeeABLE [37] | | ✓ | | ✓ | |
| RBI [64] | ✓ | | ✓ | | |
| Plug-and-Play [83] | | ✓ | | ✓ | |
| **Ours** | ✓ | ✓ | ✓ | ✓ | ✓ |

## 2 Related Work

**Classic Deepfake Detection.** Classic deepfake detection can be categorized into multiple aspects: On the one hand, typical deepfake image detection methods encompass frequency-domain analysis [56, 49], identity information [17, 9, 69], contrastive learning [23, 87], network structure improvement [88, 17], watermark [85, 72], robustness [38, 51], interpretability [16, 62] and so forth. On the other hand, deepfake video detection methods include innovative video representation formats [78], self-supervised learning [2, 27, 92], anomaly detection [19, 20], biological signal [76, 8], multi-modal based [54]. In addition to the above two aspects, the the remainder of the work involves forgery localization [51, 26], multi task learning [51, 86], adversarial attack [44], and so on.

**Deepfake Detectors Based on Data Synthesis.** A notable approach involves synthesizing pseudo-fake faces during training to enhance generalization capability. Early works like Face X-ray [40] pioneered blending-based augmentation by fusing facial regions from different identities to simulate forgery boundaries. Subsequent methods improved upon this paradigm through different artifact simulation strategies: SLADD [3] focused on local adversarial perturbations, SBI [63] proposed self-blending to amplify facial inconsistencies, SeeABLE [37] which proposes fine-grained region-specific blending, and Plug-and-Play [83] employs facial feature masks rather than full-face manipulation for enhanced face synthesis precision. However, as highlighted in Table 1, existing methods predominantly focus on **isolated artifact types** (Mainly focused on FIA) and **single-scale modifications** (global or local), fundamentally limiting their ability to capture the artifacts interplay inherent in real deepfakes. The paradigm of this data augmentation method can be represented as:

$$I_F = M \odot I_t + (1 - M) \odot I_s, \tag{1}$$

where, $I_t$ represents the target face, $I_s$ represents the source face, and $M$ represents the mask. Our proposed FIA-USA method injects new vitality into this paradigm by generating multiple types of masks, introducing USA to $I_t$, as well as random combination of artifacts.

## 3 Method

In this section, we begin by outlining two distinct and prevalent types of forgery artifacts found in deepfakes: **Face Inconsistency Artifacts (FIA)** and **Up-Sampling Artifacts (USA)**. We then detail the construction of data augmentation techniques (**FIA-USA**) designed to enable the model to learn to

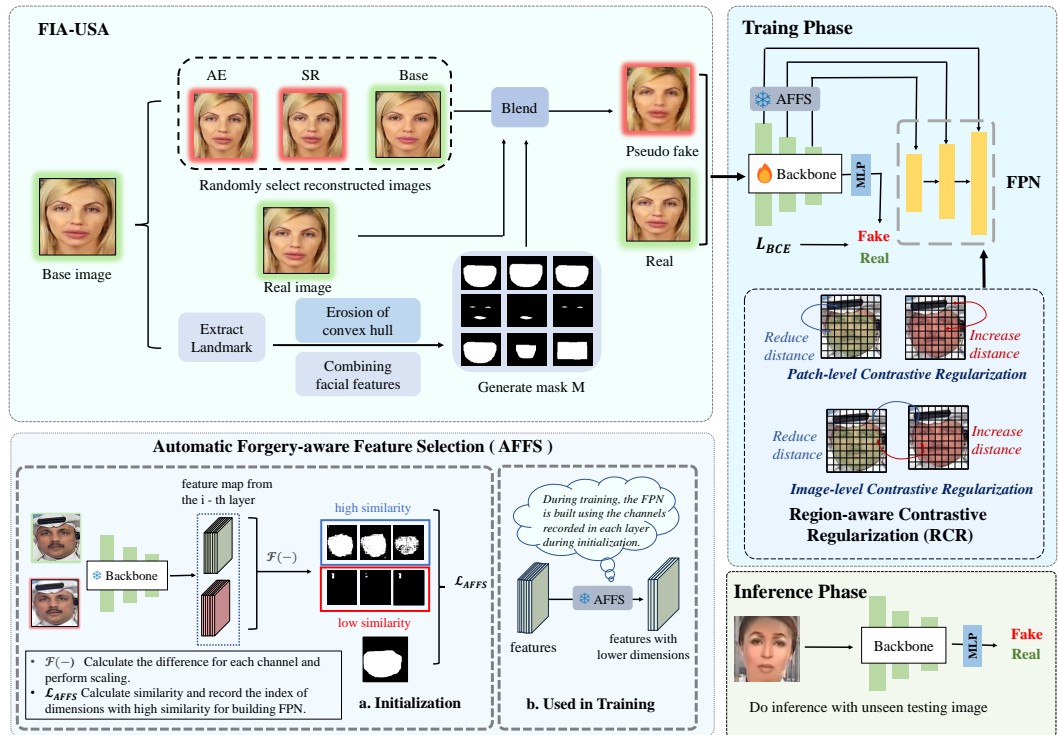

Figure 2: The overall framework of our proposed method consists of: 1) **FIA-USA** enhances the capability of standard classifiers to effectively detect and generalize to unknown deepfake techniques by generating pseudo-fake data containing Face Inconsistency Artifacts (FIA) and Up-Sampling Artifacts (USA). 2) **Region-aware Contrastive Regularization (RCR)** enables the model to focus simultaneously on forged boundaries and spatial artifacts through the contrastive regularization of features from forged and real regions. 3) **Automatic Forgery-aware Feature Selection (AFFS)** achieves efficient feature selection by assessing the similarity between the feature maps of each channel in each layer and the mask.

detect these artifacts concurrently. Subsequently, we present the **Automatic Forgery-aware Feature Selection (AFFS)** and **Region-aware Contrastive Regularization (RCR)**.

### 3.1 FIA-USA.

**Face Inconsistency Artifacts (FIA).** As depicted in Figure 1, Face Inconsistency Artifacts (FIA) frequently occur in deepfakes due to the inherent difficulty in rendering intricate details, which often results in discrepancies between intricate facial features and the more uniform surrounding regions. In essence, until deepfake technology advances to the point where it can flawlessly replicate real faces, these artifacts are likely to persist. We explore the origins of Face Inconsistency Artifacts by categorizing deepfake techniques into two domains: *Full Face Synthesis* and *Face Swapping*. In deepfake-based full face synthesis, Face Inconsistency Artifacts inherently persist regardless of the generator's capabilities, while their severity directly correlates with the model's performance. For face swapping, these artifacts arise from two primary sources: **1).** the efficacy of the generator. **2).** the blending boundary created during the face swapping's blending procedure. Furthermore, we can categorize face swapping into two main types based on the forged area: *a. macro-editing* realize face swapping through 4 or 81 facial keypoints. and *b. micro-editing*, which enables granular control over individual facial features. **In section B of the appendix, we provide a detailed description of the classification of forgery techniques and their differences.**

**Up-Sampling Artifacts (USA).** These artifacts consistently present in synthetic facial regions originate from upsampling operations within the generator's decoder architecture, where interpolation processes inevitably introduce characteristic distortion patterns, with the extent of USA (Up-Sampling Artifacts) introduced varying significantly across different generator implementations. To investigate

more universal up-sampling artifacts, we categorize forgery techniques into two types: One category generates images of forged regions that are as clear and free of blur as the original image. And another type result in blurred tampered areas, requiring additional facial super-resolution models (SR) to process the blurred areas.

**FIA-USA.** While FIA and USA represent two fundamental artifact categories in deepfakes, conventional data augmentation methods predominantly focus on simulating isolated forgery traces, thereby hindering detection models from effectively learning cross-domain forgery patterns. To bridge this critical gap, we propose FIA-USA—a dual-artifact collaborative enhancement framework engineered for universal detection. Our framework systematically emulates the forgery process through a three-pronged methodology:

**1. Multi-Type Mask Synthesis(MTMS):** The Multi-Type Mask Synthesis module aims to comprehensively model Face Inconsistency Artifacts (FIA) through a hierarchical mask generation strategy, covering both macro-editing boundaries and micro-editing traces. This process involves two complementary approaches:
*a. Macro-Editing Mask.* We generate facial contours through two configurations: (1) a high-precision mode using 81 facial keypoints to compute detailed convex hull boundaries, and (2) a rectangular approximation mode derived from 4 keypoints. The initial mask $\mathbb{M}$ is formulated as:

$$\mathbb{M} = \mathcal{H}(K_n), n \in \{4, 81\} \tag{2}$$

where $\mathcal{H}(\cdot)$ denotes convex hull computation and $K_n$ represents the keypoints set. As same as [63], the initial mask $\mathbb{M}$ subsequently undergoes erosion or dilation by applying two Gaussian filters with distinct kernel sizes, where erosion occurs when the kernel of the first filter is larger than that of the second, while dilation manifests when the first kernel is comparatively smaller.
*b. Micro-Editing Masks.* Given a set of 81 facial keypoints $K_{81} = \{k_1, k_2, \ldots, k_{81}\}$, we first extract two subsets: eyes-specific keypoints $K_{eyes} \subset K$ and mouth-specific keypoints $K_{mouth} \subset K$. Using convex hull computation $\mathcal{H}(\cdot)$, we derive two binary masks: the eye mask $M_e = \mathcal{H}(K_{eyes})$ and the mouth mask $M_m = \mathcal{H}(K_{mouth})$. These masks are then logically combined through union operations $\cup$ to generate three distinct facial regions: (1) $M = M_e$, (2) $M = M_m$, and (3) $M = M_e \cup M_m$ (combined mouth and eyes). Since other facial regions such as eyebrows and nose are rarely targeted in localized facial manipulations (e.g., attribute editing, expression synthesis), these features were intentionally excluded from our mask generation framework to prioritize regions most susceptible to forgery (eyes and mouth).

**2. Multi-Modal Reconstruction(MMR):** We employ two complementary reconstruction paradigms to reconstruct the target face($I_t$) in Formula 1, thereby systematically simulating the Up-Sampling Artifacts (USA) inherent in diverse deepfake generators: *a. Autoencoder (AE) Reconstruction:* Reconstruct $I_t$ through AE [57] to simulate the excessive dependency relationship between adjacent pixels [65] caused by upsampling in GAN / Diffusion models. *b. Super-Resolution (SR) Reconstruction:* Applies SR models [71] to upsample $I_t$, thereby replicating the characteristic artifacts inherent in deepfake post-processing pipelines. Specifically, given an input face image $I$, we generate reconstructed versions $I_{AE} = AE(I)$ and $I_{SR} = SR(I)$.

**3. Random Artifact Combination(RAC):** To ensure comprehensive coverage of artifact interactions, we blend original images(source face) with reconstructed variants(target face) $(I_{AE}, I_{SR}, I)$ using mask $M$, based on Equation 1. More specifically, given a facial image$I_s$, first MTMS generates multiple masks $M$ with equal probability, then MMR generates a reconstructed images $(I_{AE}, I_{SR})$, and finally RAC blend the $I$ with randomly selected reconstruction variants $I_t$ from $(I_{AE}, I_{SR}, I)$ using a randomly selected mask $M$. Before blending, $I_s$ and $I_t$ will undergo data augmentation(change color tone, saturation, contrast, blur, etc) to enhance the inconsistency of statistical information.
As shown in Table 1, compared to existing methods, FIA-USA has achieved collaborative artifact enhancement across local / global, facial / regional levels. Please refer to Algorithm 1 for the complete algorithmic process.

## 3.2 Loss Function

### 3.2.1 Automatic Forgery-aware Feature Selection

While FIA-USA generates discriminative training samples through dual-artifact augmentation, the inherent feature redundancy in Feature Pyramid Networks (FPN) [45] may drive models to focus on

**Algorithm 1** Pseudocode for FIA-USA

---

**Input**: Base image $I$ of size $(H, W, 3)$, facial landmarks $L$ of size $(81, 2)$
**Output**: Image $I$ with FIA and USA

1: **def** $\mathcal{T}(I)$ **:**                                      ▷ Source-Target Augmentation
2:      $I \leftarrow$ ColorTransform$(I)$
3:      $I \leftarrow$ FrequencyTransform$(I)$
4:      **return** $I$
5: **def** $Recon(I)$ **:**                                      ▷ Reconstruct the source image
6:      **if** Uniform$(\min = 0, \max = 1) \in [0, 0.3]$ **:**
7:          $I_t \leftarrow AE(I)$
8:      **else if** Uniform$(\min = 0, \max = 1) \in (0.3, 0.5]$ **:**
9:          $I_t \leftarrow SR(I)$
10:      **else**
11:          $I_t \leftarrow I$
12:      **return** $I$
13: $I_t, I_s \leftarrow Recon(I), I$
14: **if** Uniform$(\min = 0, \max = 1) < 0.5$ **:**
15:      $I_s, I_t \leftarrow \mathcal{T}(I_s), I_t$
16: **else :**
17:      $I_s, I_t \leftarrow I_s, \mathcal{T}(I_t)$
18: $M \leftarrow$ CombineFacialFeatures$(L)$ or ConvexHull$(L)$
19: $I_{PF} \leftarrow I_s \odot M + I_t \odot (1 - M)$

---

non-critical forgery patterns, thereby compromising detection generalization. This limitation stems from FPN's naive aggregation of all channel-wise features without adaptively emphasizing FIA/USA-correlated representations. To address this limitation, we propose Automatic Forgery-aware Feature Selection (AFFS), a statistically-driven feature compression paradigm that quantifies channel-wise sensitivity to forgery regions, thereby dynamically constructing lightweight yet discriminative feature pyramids.

Given a pretrained backbone network $\phi$, let $f_i = \phi_i(I) \in \mathbb{R}^{W_i * H_i * C_i}$ denote the feature map from the $i$-th layer, where $I \in \mathbb{R}^{H * W * 3}$ represents the input image. To identify artifact-sensitive feature dimensions, we leverage pseudo-fake pairs $\{R_n, F_n, M_n\}_{n=1}^N$ generated by FIA-USA and compute the normalized response discrepancy between real and forged regions. For the $k$-th channel $f_i^k \in \mathbb{R}^{W_i * H_i}$ in layer $i$, the selection criterion is formulated as:

$$\mathcal{L}_{AFFS}^{i,k} = \frac{1}{N} \sum_{n=1}^N \|\mathcal{F}(f_i^k(R_n) - (f_i^k(F_n)) - M_n\|_2^2 \tag{3}$$

where $\mathcal{F}$ denotes feature normalization and spatial interpolation to align $f_i^k$ with mask $M_n$. Channels are ranked by ascending $\mathcal{L}_{AFFS}^{i,k}$ and the top-$m_i$ channels with minimal errors are retained to form a compressed feature map $f_i' \in \mathbb{R}^{W_i * H_i * m_i}$ for FPN construction. **AFFS leverages a pre-trained model as initialization, operating as a supervised feature ranking mechanism that reduces feature redundancy.**

### 3.2.2 Region-aware Contrastive Regularization

Building upon the artifact-sensitive features selected by AFFS, we design Region-aware Contrastive Regularization (RCR) to explicitly model the divergence between forged and authentic regions through multi-granularity contrastive learning. Given the Feature Pyramid Network (FPN) $P$, we denote the feature maps of real and fake faces as $H_R = P(\phi(R))$, $H_F = P(\phi(F))$. For $H_R$, the areas inside and outside $M$ are real, while for $H_F$, the areas inside $M$ are manipulated and the areas outside $M$ are real. We define the region within $M$ as $H_R^{in}$, $H_F^{in}$ and the region outside $M$ as $H_R^{out}$, $H_F^{out}$. RCR operates on the refined feature pyramid from AFFS, implementing multi-granularity contrast through:

**Patch-level Contrastive Regularization.** For real faces, the features $f$ of $H_R^{in}$ and $H_R^{out}$ are consistent, thus we aim to minimize the distance between them. Conversely, for fake faces, the features $f$ of $H_F^{in}$ and $H_F^{out}$ are distinct, thus we aim to maximize the distance between them. Therefore, the loss function of **Patch-level Contrastive Regularization (PCR)** can be formulated as:

$$\mathcal{L}_1 = -\log \frac{\sum\limits_{f_R \in H_R} e^{\delta(f_R^{in}, f_R^{out})/\tau}}{\sum\limits_{f_R \in H_R} e^{\delta(f_R^{in}, f_R^{out})/\tau} + \sum\limits_{f_F \in H_F} e^{\delta(f_F^{in}, f_F^{out})/\tau}}, \tag{4}$$

where $f^i$ represents a pixel feature, $\tau$ is a temperature parameter and $\delta(\cdot)$ is the normalized cosine similarity between two features, as:

$$\delta(f_1, f_2) = \sum_{f_1^i \in f_1} \sum_{f_2^i \in f_2} \frac{f_1^i}{\|f_1^i\|} \cdot \frac{f_2^i}{\|f_2^i\|}, \tag{5}$$

**Image-level Contrastive Regularization.** Since both the region of real face and fake face outside $M$ are real, we aim to minimize the distance between them and maximize the distance between the fake face and the real face in the area inside $M$. Therefore, the loss function for **Image-level Contrastive Regularization (ICR)** can be formulated as:

$$\mathcal{L}_2 = -\log \frac{\sum\limits_{f_{out} \in H^{out}} e^{\delta(f_R^{out}, f_F^{out})/\tau}}{\sum\limits_{f^{out} \in H_{out}} e^{\delta(f_R^{out}, f_F^{out})/\tau} + \sum\limits_{f^{in} \in H^{in}} e^{\delta(f_R^{in}, f_F^{in})/\tau}}. \tag{6}$$

**Overall Loss.** The network is optimized using the following loss:

$$\mathcal{L} = \lambda_1 \mathcal{L}_{BCE} + \lambda_2 \mathcal{L}_1 + \lambda_3 \mathcal{L}_2, \tag{7}$$

where $\mathcal{L}_{BCE}$ denotes the cross-entropy classification loss. $\mathcal{L}_{BCE}$, $\mathcal{L}_1$ and $\mathcal{L}_2$ are weighted by the hyperparameters $\lambda_1$, $\lambda_2$ and $\lambda_3$, respectively.

## 4 Experiments

### 4.1 Settings

**Datasets.** To evaluate the effectiveness of our proposed method, we conducted extensive experiments on **seven widely-adopted benchmark datasets** spanning both classical facial manipulation paradigms and emerging generative deepfake architectures.**a. Traditional Deepfake Datasets:** 1. FaceForensics++ (FF++) [59], 2. Deepfake Detection (DFD) [12], 3. Deepfake Detection Challenge (DFDC) [15], 4. preview version of DFDC (DFDCP) [14], and 5. CelebDF (CDF) [43]. FF++ comprises 1,000 original videos and 4,000 fake videos forged by four manipulation methods, namely, Deepfakes(DF) [13], Face2Face(F2F) [68], FaceSwap(FS) [18], and NeuralTextures(NT) [67]. FF++ offers three levels of compression: raw, lightly compressed (c23), and heavily compressed (c40), **under storage constraints, our implementation adopts the FF++\_c23 (including DFD) with training conducted following the SBI protocol [63], employing exclusively real facial samples from FF++\_c23 subset.** Although many previous studies have utilized the same dataset for both training and testing, the preprocessing and experimental configurations can differ, making fair comparisons difficult. Therefore, in addition to testing on the raw data of the aforementioned datasets, we also performed generalization assessments on the unified new benchmark for traditional deepfakes, (DeepfakeBench) [82]. **b. Generative Deepfake Datasets.** 6. Diffusion Facial Forgery (DiFF) [5] contains over 500,000 images synthesized using 13 different generation methods under four conditions: Text to Image (T2I), Image to Image (I2I), Face Swapping (FS) and Face Editing (FE). **Text to Image** encompasses four generation methods: *Midjourney* [50], *Stable Diffusion XL (SDXL)* [55], *Free-DoM\_T* [84], and *HPS* [77]. **Image to Image** comprises *Low Rank Adaptation(LoRA)* [31], *DreamBooth* [60], *SDXL Refiner* [55], and *Free-DoM\_I*. **Face Swapping** includes *DiffFace* [36] and *DCFace* [36]. **Face Editing** comprises *Imagic* [35], *Cycle Diffusion(CycleDiff)* [75], and *Collaborative Diffusion(CoDiff)* [34]. **7. DF40 [80] encompasses 40 state-of-the-art deepfake generation methodologies spanning four core categories: face swapping, facial reenactment, full-face synthesis, and advanced facial manipulation.** Significantly surpassing existing benchmarks in both scope and volume, DF40 integrates cutting-edge generative architectures from 2024 alongside widely adopted commercial software solutions.
**Evaluation Metrics.** We report the Frame-Level Area Under Curve (AUC) metric on the DeepfakeBench and DIFF dataset, report the Video-Level Area Under Curve (AUC) metric on the traditional deepfake datasets, to compare our proposed method with prior works.
**Implementation Details.** We adopt EfficientNetB4 [66] as the backbone network architecture (We also

Table 2: **Cross-dataset Evaluations:** Cross-dataset evaluations were conducted using the **Frame-level AUC**. All experiments were trained on the **c23 version of FF++** and tested on other datasets. * indicates the results are cited from [10] and † indicates our reproduction results using the checkpoints provided by the authors, otherwise, the results are from DeepfakeBench [82].

| Detector | Backbone | Venues | CDF-v1 | CDF-v2 | DFD | DFDCP | Avg. |
|---|---|---|---|---|---|---|---|
| FWA [41] | Xception | CVPRW'18 | 0.790 | 0.668 | 0.740 | 0.638 | 0.714 |
| CapsuleNet [52] | Capsule | ICASSP'19 | 0.791 | 0.747 | 0.684 | 0.657 | 0.720 |
| CNN-Aug [29] | ResNet | CVPR'20 | 0.742 | 0.703 | 0.646 | 0.617 | 0.677 |
| Face X-ray [40] | HRNet | CVPR'20 | 0.709 | 0.679 | 0.766 | 0.694 | 0.712 |
| FFD [11] | Xception | CVPR'20 | 0.784 | 0.744 | 0.802 | 0.743 | 0.768 |
| F3Net [56] | Xception | ECCV'20 | 0.777 | 0.798 | 0.702 | 0.735 | 0.749 |
| SPSL [47] | Xception | CVPR'20 | 0.815 | 0.765 | 0.812 | 0.741 | 0.783 |
| SRM [49] | Xception | CVPR'21 | 0.793 | 0.755 | 0.812 | 0.741 | 0.775 |
| CORE [53] | Xception | CVPRW'22 | 0.780 | 0.743 | 0.802 | 0.734 | 0.764 |
| Recce [2] | Designed | CVPR'22 | 0.768 | 0.732 | 0.812 | 0.734 | 0.761 |
| SBI* [62] | EfficientNet-B4 | CVPR'22 | - | 0.813 | 0.774 | 0.799 | - |
| UCF [81] | Xception | ICCV'23 | 0.779 | 0.753 | 0.807 | 0.759 | 0.774 |
| ED* [1] | ResNet-34 | AAAI'24 | 0.818 | 0.864 | - | **0.851** | - |
| ProDet† [6] | EfficientNet-B4 | NeurIPS'24 | **0.909** | 0.842 | 0.848 | 0.774 | 0.843 |
| LSDA* [79] | EfficientNet-B4 | CVPR'24 | 0.867 | 0.830 | **0.880** | 0.815 | 0.848 |
| Forensic-Adapter* [10] | CLIP (ViT-B/16) | CVPR'25 | - | 0.837 | - | 0.799 | - |
| Ours | EfficientNet-B4 | - | 0.901 | **0.867** | 0.821 | 0.818 | **0.852** |

investigate alternative network architectures and their respective outcomes), trained for 50 epochs using the SAM optimizer [21] with a batch size of 12 and initial learning rate of 0.001. For video processing, each input is uniformly sampled into 32 frames during both training and inference phases. Our data augmentation pipeline combines the proposed FIA-USA strategy with conventional techniques including RandomHorizontalFlip, RandomCutOut, and AddGaussianNoise. The loss coefficients $\lambda_1, \lambda_2, \lambda_3$ are empirically set to 1, 2.5, and 0.25 respectively (We also explored the impact of other variants on the detection results), with the temperature parameter $\tau$ fixed at 0.7. All experiments were conducted on a single NVIDIA 3090.

## 4.2 Generalization Evaluation

### 4.2.1 Traditional Deepfake Datasets.

We first compare our method with previous work at the **frame level**. To ensure the fairness of the experiment, we conducted the experiment on the DeepfakeBench, and adhered to the data preprocessing and experimental settings provided by them. For previous work, we utilized the experiment data provided by DeepfakeBench. As shown in Table 2, our method outperforms other methods on the majority of deepfake datasets and competes with state-of-the-art methods on the CDF-v1 dataset. In addition to comparing at the frame-level, we also compared our method at the **video-level** with state-of-the-art detection algorithms, including various data augmentation methods. The comparison results, as shown in Table 3, strongly demonstrate the effective generalization of our method.

Table 3: **Comparison with SOTA methods using the Video-Level AUC.**

| Model | Venues | CDF-v2 | DFDC | DFDCP |
|---|---|---|---|---|
| Face X-Ray [40] | CVPR'20 | - | - | 0.711 |
| LipForensics [28] | CVPR'21 | 0.824 | 0.735 | - |
| FTCN [90] | ICCV'21 | 0.869 | - | 0.740 |
| PCL + I2G [89] | ICCV'21 | 0.900 | 0.675 | 0.743 |
| HCIL [24] | ECCV'22 | 0.790 | 0.692 | - |
| ICT [17] | CVPR'22 | 0.857 | - | - |
| SBI [63] | CVPR'22 | 0.928 | 0.719 | 0.855 |
| AltFreezing [74] | CVPR'23 | 0.895 | - | - |
| SAM [7] | CVPR'24 | 0.890 | - | - |
| LSDA [79] | CVPR'24 | 0.911 | **0.770** | - |
| CFM [48] | TIFS'24 | 0.897 | - | 0.802 |
| LAA-Net† [51] | CVPR'24 | 0.840 | - | 0.741 |
| Ours | - | **0.941** | 0.732 | **0.866** |

Table 4: **Comparison with universal deepfake detection methods using the Frame-Level AUC on the DiFF dataset.** † indicates models that are designed for deepfake detection, while ‡ signifies models intended for generated image detection.

| Method | Test Subset | | | |
|---|---|---|---|---|
| | T2I | I2I | FS | FE |
| Xception† [59] | 62.43 | 56.83 | 85.97 | 58.64 |
| F3-Net† [56] | 66.87 | 67.64 | 81.01 | 60.60 |
| EfficientNet† [66] | 74.12 | 57.27 | 82.11 | 57.20 |
| DIRE‡ [73] | 44.22 | 64.64 | 84.98 | 57.72 |
| SBI† [63] | 80.20 | 80.40 | 85.08 | 68.79 |
| Ours | **86.05** | **84.95** | **89.42** | **72.73** |

### 4.2.2 Generative Deepfake Datasets

Our evaluation on DIFF and DF40 datasets demonstrates the superior generalization capabilities of our architecture compared to state-of-the-art detection methods. Comprehensive metrics are detailed in Table 4 and Table 5. As shown in Table 5, our method achieves an average AUC of **87.8%**, surpassing RECCE by **9.7%** and outperforming SBI by **23.4%**, and enables **10%** higher AUC than LSDA's latent space augmentation on

Table 5: **Comparative Analysis of Detection Performance Between the Proposed Method and State-of-the-Art Approaches on Six Representative Face Swapping Forgery Types in DF40 [80].**

| Method | Venues | uniface | e4s | facedancer | fsgan | inswap | simswap | Avg. |
|---|---|---|---|---|---|---|---|---|
| RECCE [2] | CVPR 2022 | 84.2 | 65.2 | 78.3 | 88.4 | 79.5 | 73.0 | 78.1 |
| SBI [63] | CVPR 2022 | 64.4 | 69.0 | 44.7 | 87.9 | 63.3 | 56.8 | 64.4 |
| CORE [53] | CVPRW 2022 | 81.7 | 63.4 | 71.7 | **91.1** | 79.4 | 69.3 | 76.1 |
| IID [33] | CVPR 2023 | 79.5 | 71.0 | 79.0 | 86.4 | 74.4 | 64.0 | 75.7 |
| UCF [81] | ICCV 2023 | 78.7 | 69.2 | 80.0 | 88.1 | 76.8 | 64.9 | 77.5 |
| LSDA [79] | CVPR 2024 | 85.4 | 68.4 | 75.9 | 83.2 | 81.0 | 72.7 | 77.8 |
| CDFA [46] | ECCV 2024 | 76.5 | 67.4 | 75.4 | 84.8 | 72.0 | 76.1 | 75.9 |
| ProgressiveDet [6] | NeurIPS 2024 | 84.5 | 71.0 | 73.6 | 86.5 | 78.8 | 77.8 | 78.7 |
| Ours | - | **91.8** | **87.5** | **83.0** | 86.3 | **87.4** | **91.0** | **87.8** |

(a) **Ablation Study of Method Components.** We report the Video-Level AUC for traditional deepfake datasets and the Frame-Level AUC for generative deepfake datasets. SR signifies the process of super-resolution, and AE refers to the reconstruction achieved through autoencoders.

| Setting | Test Dataset | | | | Avg. |
|---|---|---|---|---|---|
| | Traditional | | Generative | | |
| | CDF-v2 | DFDCP | FS | FE | |
| w/o MTMS | 91.99 | 87.06 | 88.00 | 71.93 | 84.74 |
| w/o SR | 93.25 | 86.81 | 80.73 | 61.07 | 80.46 |
| w/o AE | 92.53 | 86.37 | 88.67 | 70.31 | 84.47 |
| w/o FIA-USA | 93.18 | 89.45 | 80.68 | 62.99 | 81.57 |
| w/o PCR | 90.12 | 86.13 | 89.61 | 66.87 | 83.18 |
| w/o ICR | 90.85 | 85.00 | 89.95 | 75.01 | 85.20 |
| w/o RCR | 90.14 | 86.13 | 89.61 | 66.87 | 83.18 |
| w/o AFFS | 94.70 | 86.30 | 88.33 | 70.62 | 84.98 |
| Ours | 94.10 | 86.66 | 89.42 | 72.73 | 85.72 |

(b) **Ablation Study on Model Architectures.**

| Model | Test Dataset | | | | Avg. |
|---|---|---|---|---|---|
| | Traditional | | Generative | | |
| | CDF-v2 | DFDCP | FS | FE | |
| Res50 | 82.45 | 79.14 | 68.82 | 58.59 | 72.25 |
| Res101 | 85.51 | 82.35 | 71.26 | 60.44 | 74.89 |
| Effb1 | 91.88 | 82.30 | 83.73 | 70.69 | 82.15 |
| Effb4 | 94.10 | 86.66 | 89.42 | 72.73 | **85.72** |

(c) **Experiment on Hyperparameter Configuration.**

| $\lambda_1, \lambda_2, \lambda_3$ | Test Dataset | | | | Avg. |
|---|---|---|---|---|---|
| | Traditional | | Generative | | |
| | CDF-v2 | DFDCP | FS | FE | |
| 1, 1, 1 | 90.86 | 85.01 | 89.91 | 70.19 | 83.99 |
| 1, 1, 0.25 | 91.88 | 86.01 | 89.91 | 72.19 | 85.00 |
| 1, 2.5, 1 | 91.82 | 86.00 | 89.95 | 71.01 | 84.70 |
| 1, 0.25, 2.5 | 91.11 | 85.59 | 89.61 | 70.92 | 84.30 |
| 1, 2.5, 0.25 | 94.10 | 86.66 | 89.42 | 72.73 | **85.72** |

Table 6: **Ablation experiment.** We conducted ablation experiments on method components, model frameworks, and hyperparameter configuration. The best results are presented in bold.

average. The results notably contradict recent findings [79] about RGB-based methods' limitations against generative deepfakes, proving that properly designed RGB-level artifacts retain critical discriminative signals. **For comprehensive experimental results on forgery techniques supported by the DF40 and DIFF, please refer to the extended analysis in Appendix Section G.**

### 4.2.3 Robustness

In Figure 3, we assessed the influence of different levels of random perturbations on the detection performance. We quantified the effects of different levels of Gaussian Blur, Block Wise, Change Contrast, and JPEG Compression on the detectors. Notably, to gauge accuracy, we employed the reduction rate of Video-Level AUC to assess the robustness of the detector to different degrees of disturbance, which could be expressed as $R_{AUC} = (AUC - AUC_{raw})/AUC_{raw}$, where $AUC_{raw}$ refers to no perturbations. As shown in the Figure 3, our method exhibits superior robustness compared to other methods.

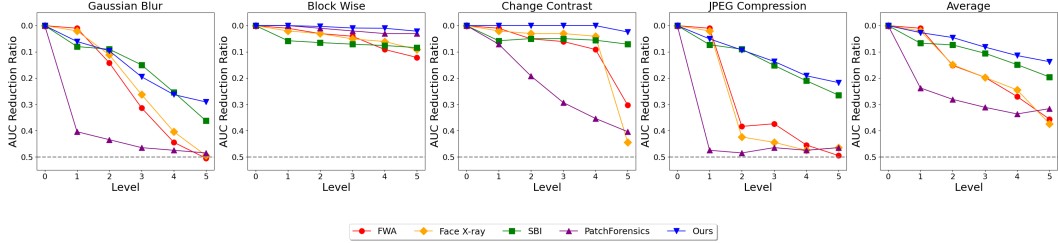

Figure 3: **Robustness to Unseen Perturbations.** We reported the Video-Level **AUC Reduction Ratio** for four specific types of perturbations at five different levels.

### 4.3 Ablation Study

**The impact of method components.** To systematically evaluate the contribution of each component in our framework, we conducted comprehensive ablation studies across seven benchmark datasets. As detailed in Table 6a, our analysis focuses on three core innovations: (1) the FIA-USA augmentation mechanism, (2) Automatic Forgery-aware Feature Selection (AFFS), and (3) Region-aware Contrastive Regularization (RCR). We adopt video-level AUC for traditional deepfakes and frame-level AUC for generative deepfakes. The empirical results demonstrate progressive performance degradation when removing individual components, with the complete framework achieving optimal detection accuracy . The combination of FIA-USA, AFFS, and RCR yields optimal performance. As intended in our design, AFFS and RCR are crucial for fully leveraging the potential of FIA-USA. Removing any component degrades performance, with the absence of FIA-USA having the most significant impact (a 4.15% decrease). It is important to clarify that "w/o FIA-USA" specifically refers to using only the data augmentation proposed by SBI. Furthermore, RCR exerts a stronger influence on FIA-USA's effectiveness compared to AFFS.

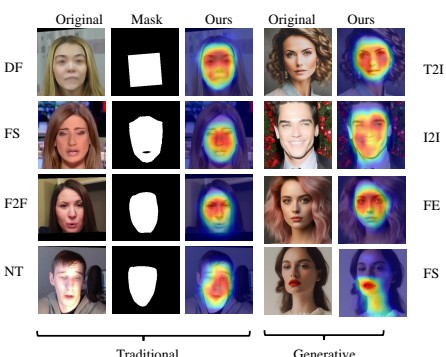

Figure 4: **GradCAM visualization of fake samples across various deepfake datasets.**

The ablation specifically targeting FIA-USA (components within it) reveals: Various data augmentation techniques contribute differently. SR and AE are shown to be effective for detecting generative forgeries. Conversely, MTMS appears to suppress performance on this specific forgery type. This aligns with our theoretical analysis: generative forgeries constitute global manipulations that produce fewer Face Inconsistency Artifacts (FIA) but primarily exhibit Up-Sampling Artifacts (USA). As seen from MTMS's performance on traditional forgeries, increasing the diversity of FIA proves beneficial for detecting face swapping forgery.

Regarding the RCR ablation: The results align well with our chosen hyperparameters. They indicate that the PCR contributes substantially more to the model's performance than the ICR. This observation is consistent with the hyperparameter sensitivity analysis presented in Table 6c: increasing the loss weight for PCR while decreasing that for ICR leads to significant performance gains. Conversely, increasing the ICR weight while reducing the PCR weight results in only marginal improvements.

**The impact of model framework.** Our architectural analysis in Table 6b systematically benchmarks detection performance across backbone networks, quantitatively comparing the effects of EfficientNet [66], ResNet [29], and different depth configurations on detection performance, demonstrates the compatibility of our method with various network frameworks, while also illustrating the critical impact of model parameter size and architectural design on detector performance. This highlights the ongoing importance of developing more robust and specialized deep learning architectures for counterfeit detection tasks.

**Impact of hyperparameters.** We also examined the effect of hyperparameters of the loss function on model performance, as detailed in the Table 6c. Our analysis of the weights revealed that increasing the weight of $\lambda_2$ relative to the weights of cross entropy enhances model performance, whereas decreasing the weight of $\lambda_3$ relative to cross entropy also improves performance, which is consistent with the ablation experiment results for RCR components in Table 6a. However, conversely, model performance will not be significantly improved.

## 5 Visualizations

Our GradCAM [91] analysis in Figure 4 demonstrates precise localization of facial forgery artifacts across both conventional and emerging generative architectures. Notably, the visualization framework not only pinpoints manipulation traces in traditional deepfakes but also effectively in state-of-the-art synthetic media, validating our method's generalization capability.

## 6 Conclusion

In conclusion, this study presents a universal framework for deepfake detection by focusing on common artifacts that span various types of face forgeries. By categorizing deepfake artifacts into Face Inconsistency Artifacts (FIA) and Up-Sampling Artifacts (USA), we enhance the generalization capability of detection models. This targeted approach allows the model to focus on detecting fundamental artifact patterns, and potentially improving its performance across diverse deepfake variations. In doing so, our framework may help address some challenges in current detection methods and could provide a promising foundation for developing more adaptable deepfake detection systems in future research.

## Acknowledgments and Disclosure of Funding

This work was supported in part by Key Science & Technology Project of Anhui Province 202423l10050033, Zhejiang Provincial Natural Science Foundation of China under Grant (No. LQN25F020023).

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

# A   Appendix

This supplementary material provides:

- Sec. B: We discussed the classification and basic process of forgery techniques.

- Sec. C: We reviewed the application of data augmentation methods in forgery detection and discussed more details about FIA-USA.

- Sec. D: We introduced the construction process of FPN.

- Sec. E: We provided a detailed introduction to AFFS was provided, along with visual evidence.

- Sec. F: We provide a visualization example of which boundary pixels were discarded by RCR.

- Sec. G: We provided test results on more forgery techniques.

- Sec. H: We provide experimental results combining FIA-USA with other state-of-the-art detection methods.

- Sec. I: We discussed the limitations of this paper.

- Sec. J: We discussed the impacts of this paper.

- Sec. K: We provided more visual examples.

# B   Face forgery techniques

In this section, we discuss the classification of facial forgery technology and the basic process of forgery technology, and at which nodes FIA and USA will be introduced.

## B.1   Classfication of facial forgery technology

According to different processing procedures and technical principles, we first divide forgery techniques into two categories:
**1).** *Full Face Synthesis:* Generate the entire image/video directly through the image / video generation model, without involving facial cropping and stitching in the basic process. Therefore, the artifacts involved in such forgery techniques mainly include USA and a small amount of FIA, where USA is the dependency between adjacent pixels in the forgery area due to the performance of the generator, and FIA is the inconsistency in the generator's ability to generate face areas with dense details and background areas with sparse details.
**2).** *Face Swapping*. Crop the original face, generate fake regions through a generator (optional), and then stitch it with the target face. Therefore, this forgery technique can be divided into two situations: a. It includes FIA and USA: FIA is mainly caused by cropping and stitching, as well as generator performance, while USA is mainly determined by generator performance. b. It only includes FIA: FIA is mainly caused by ropping and stitching. Furthermore, based on the forged area, we can divide face swapping into two parts: *Macro-editing* and *Micro-editing*. Macro-editing can be further classified into two subtypes: Editing based on 4 facial keypoints and Editing based on 81 keypoints. Micro-editing allows for precise adjustments to each facial feature.
Figure 5 provides a detailed overview of our classification concept.

## B.2   Basic Process

In Figure 6, we depict the basic process of each type of forgery technique. We can see from this that the main difference between full face synthesis and face swapping lies in whether they include face cropping and stitching operations, which leads to the emergence of a large number of FIA, and whether the background is generated by the generator. The main difference between macro-editing and micro-editing in face swapping is whether the forged area is the entire face or can finely fabricate facial features. Further, macro editing based on 81 key points is more detailed, and the edges of the forged area are more similar to the edges of a real face, while 4 key points result in the forged area being a square. This discovery is mainly due to a detailed observation of the widely used forgery technique ROPE [30] and roop [61], which provides two types of face cropping techniques.

## B.3   At what stage will FIA and USA be introduced

In Figure 7, we demonstrate which operations lead to the emergence of FIA and USA in the basic process. The observation of a phenomenon led to the refinement of our FIA-USA design.

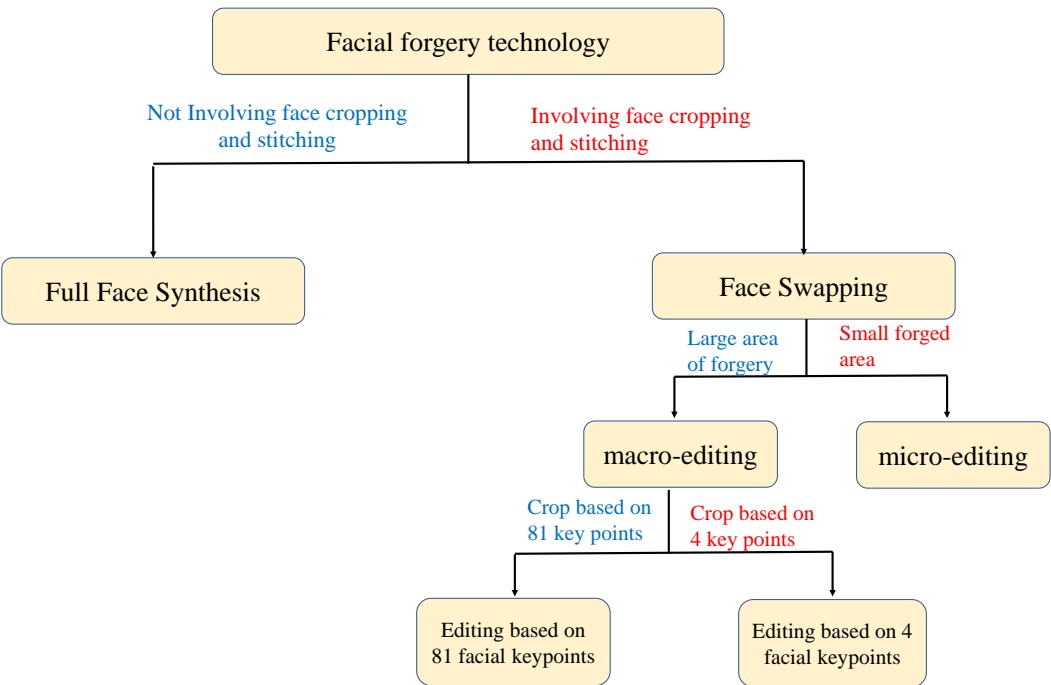

Figure 5: **We categorize forgery techniques into two main types:** *Full face synthesis* **and** *Face swapping*. Face swapping is further divided into *macro-editing* (based on 4 or 81 keypoints) and *micro-editing*.

## C   Details of FIA-USA

### C.1   Blend-based data augmentation

In Deepfake detection, a successful approach is to manually construct negative samples for training, where data augmentation in the RGB domain is based on Blend, which can be represented as follows:

$$I_F = M \odot I_t + (1 - M) \odot I_s, \tag{8}$$

Among them, $I_t$ represents the target face, $I_s$ represents the source face, and $I_F$ represents the negative sample. The earliest design [40] was to use facial similarity to find the most suitable $I_s$ for $I_t$. Later, SBI [63] proposed self-blend to improve the statistical consistency of $I_F$. There are also works [64] that separate the background and face of $I_t$, add noise to the background, and reconstruct $I_t$, using the reconstructed $I_t$ for blend. **However, to our knowledge, there has been no work so far that considers the introduction of both global and local artifacts, let alone the introduction of FIA at both the full face and regional levels while introducing USA.**

### C.2   Other types of data augmentation

In recent years, many other data augmentation methods have been proposed, such as frequency domain [39] and latent space [79]. Many studies suggest [39, 79] that compared to data augmentation in other domains, data augmentation in the RGB domain [40, 63, 64] exhibits weak robustness and limited effectiveness. However, we have demonstrated through extensive experiments that our proposed data augmentation in the RGB domain is not weaker or even better than that in other domains. **It is worth mentioning that our proposed data augmentation belongs to the image level, and we surprisingly found that it is superior to the most advanced data augmentation methods at the video level [74].**

### C.3   The detailed process of generating masks in FIA-USA

Here we have detailed the processing of generating masks in FIA-USA, as shown in Figure 8. Firstly, we divide mask generation into two types: 1) Macro editing masks and 2) Micro editing masks.
**Macro editing mask:** We follow the mask generation process in SBI [63], which includes the following steps: a. Calculate the convex hull of 81 facial keypoints to obtain the mask of facial contour b. Erosion or expansion of

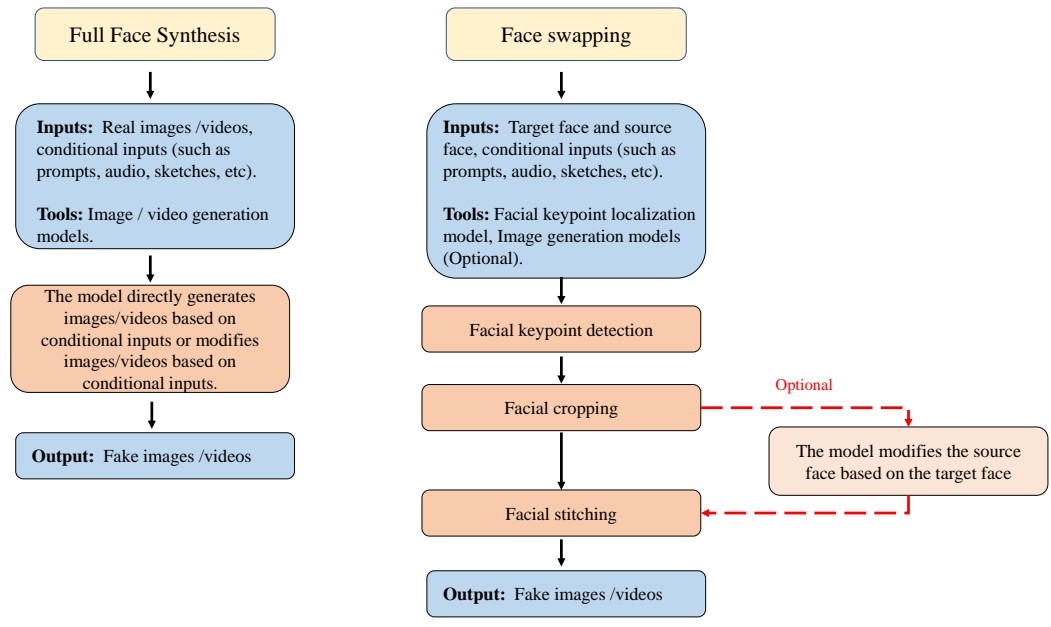

(a) Basic processes of *full face synthesis* and *face swapping*.

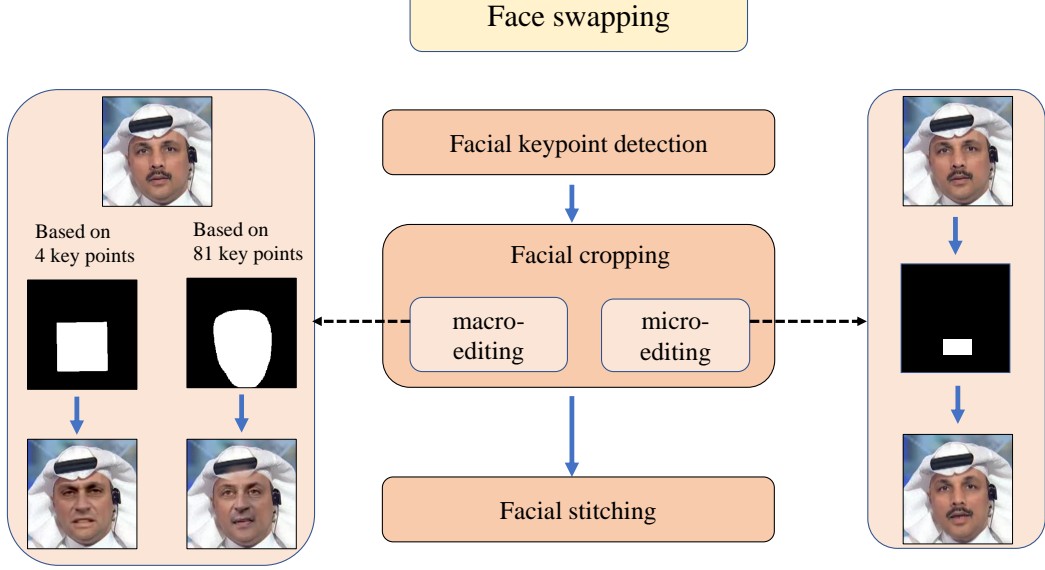

(b) Basic processes of *macro-editing (4 vs 81 keypoints)* and *micro-editing*

Figure 6: We illustrates the basic processes of forgery techniques, *full face synthesis* and *face swapping* differ in face cropping and stitching, while *macro-editing (4 vs 81 keypoints)* and *micro-editing* vary in forgery scope and detail.

the mask; Besides, we also considered the case of calculating convex hull based on 4 facial keypoints, which originated from the observation of two methods for extracting raw faces provided by the open-source deepfake project that is truly available [61, 30].

**Micro-editing masks:** The main method is to simulate the forgery process of manipulating facial details randomly by randomly combining facial features.

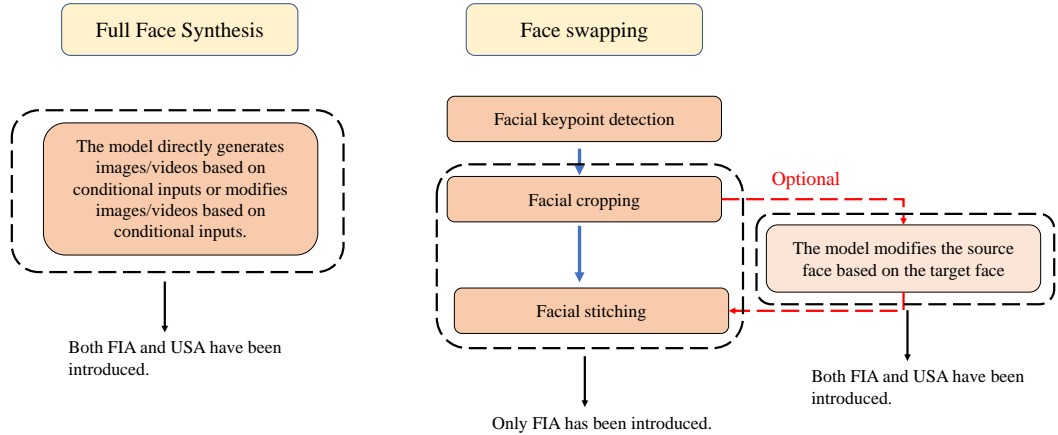

Figure 7: The operations that led to the occurrence of FIA and USA in the basic process.

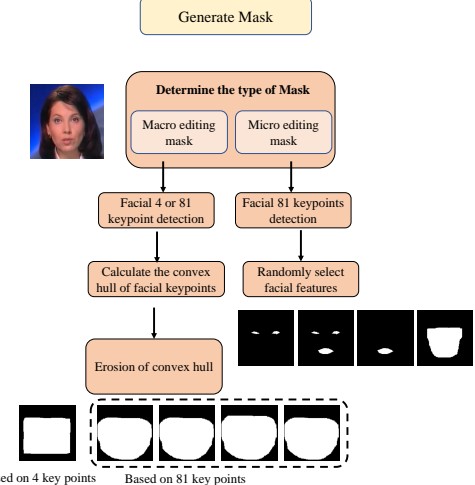

Figure 8: The generation of macro and micro-editing masks in FIA-USA, using convex hull and erosion / expansion for macro masks and random feature combination for micro masks.

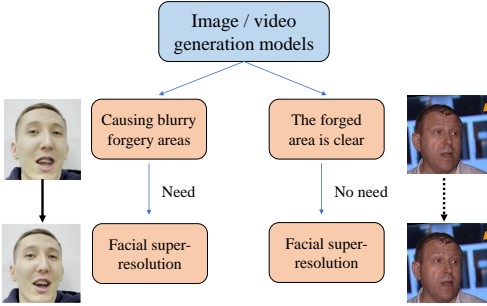

Figure 9: We divide the generation models into two categories based on whether facial super-resolution models is needed for post-processing.

## C.4 Why use Autoencoder and Super-Resolution models to reconstruct images

As shown in Figure 9, we divide the processing flow of the generative model into two categories: One category generates images of forged regions that are as clear and free of blur as the original image (using Diffusion models and GANs as the main methods, and upsampling the latent code through AE to generate images). And another

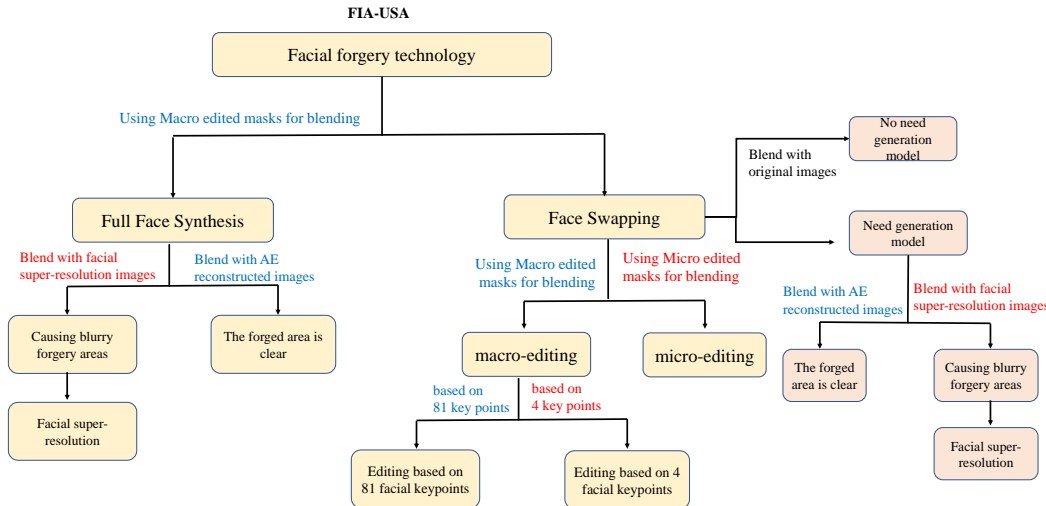

Figure 10: We demonstrate through graphical representation that our proposed FIA-USA covers all possible scenarios that may lead to the occurrence of FIA and USA.

type (especially older voice driven lip shape modification techniques) will result in blurred tampered areas, requiring additional facial super-resolution models to process the blurred areas. Therefore, we use utoencoder (AE) to reconstruct images or facial Super-Resolution (SR) to process faces, in order to approximate the USA introduced by the generator in the forged area.

## C.5 FIA-USA covers all possible situations that may occur in FIA and USA

As shown in the Figure 10, our proposed FIA-USA framework covers every possible situation that may occur in FIA and USA, which is also the most intuitive explanation why our method is effective.

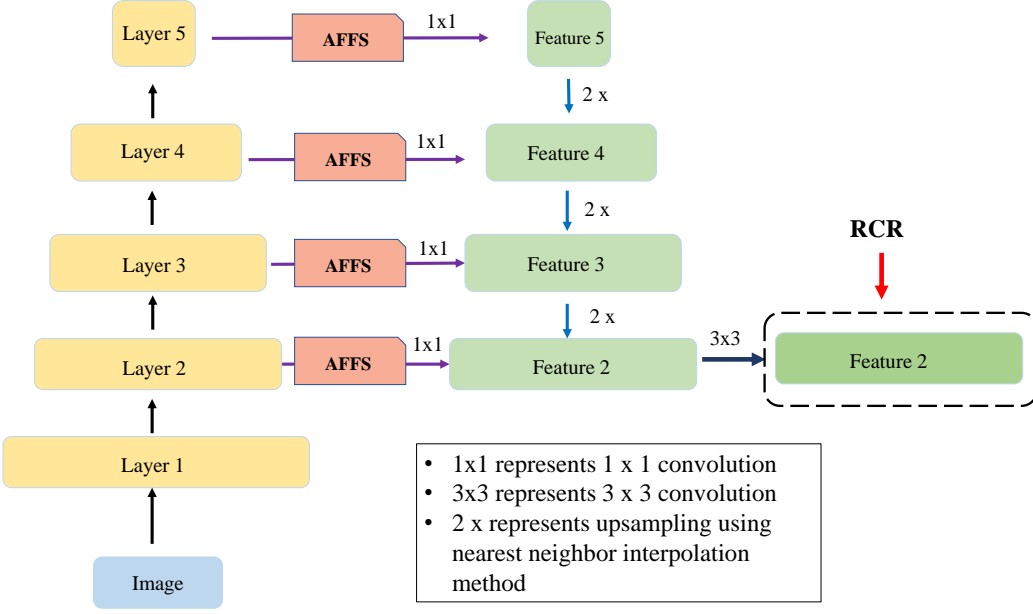

Figure 11: We illustrated the construction process of FPN.

# D Details of FPN

As shown in the Figure 11, our FPN follows the framework proposed by [45], using the output features of the 2nd, 3rd, 4th, and 5th layers of the backbone to construct an FPN, In our experimental setup, the number of feature map channels is 128 and 196, corresponding to EfficientNet [66], and Resnet [29], respectively.

# E Details of AFFS

Here we provide a detailed description of the process of AFFS and a visual explanation of its effectiveness.

## E.1 Why is AFFS proposed

Let's consider a neural network $\phi$ and an FPN $P$, as well as the process of performing RCR on the FPN's maximum resolution layer. The output of $\phi$ is a set of features $\{d_1, d_2, d_3, d_4\}$, $P$ takes this set of features as input to construct a set of feature maps with same dimensions and different resolutions, while RCR performs feature differentiation learning on the feature map with the highest resolution. One intuition is that there is a significant amount of redundancy in the output features of the $\phi$, and not every dimension of the features $d_i, i \in [1, 2, 3, 4]$ is suitable for constructing FPN for RCR. Therefore, we propose AFFS for feature dimensionality reduction of each feature $d_i$ before constructing FPN. Moreover, this feature dimensionality reduction method effectively exploits the potential advantages of our proposed FIA-USA and lays the foundation for the effective execution of RCR.

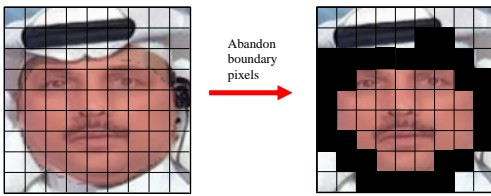

Figure 12: We provide a visual example to illustrate which pixels will be discarded.

## E.2 AFFS

1) First, let's consider multiple positive and negative sample pairs generated by FIA-USA, along with their corresponding masks $\{R_n, F_n, M_n\}_{n=1}^N$. The usual method is to directly construct an FPN using a set of features $f_i$ from a neural network $\phi$ as input, where $i$ represents the $i$-th layer of the $\phi$.

2) Next, let's discuss the initialization process of AFFS. For the output of each layer of the neural network, **we hope that the difference between the real image and the fake image after normalization is as consistent as possible with the mask $M$ used to represent forged areas.** However, we have noticed that not every channel meets this requirement, so we would like to reserve channels that meet the requirements for each layer. Therefore, for $f_i \in \mathbb{R}^{W_i * H_i * C_i}$, we calculate the following loss for each channel on a set of FIA-USA generated samples $\{R_n, F_n, M_n\}_{n=1}^N$ to obtain those excellent dimensions,

$$\mathcal{L}_{AFFS}^{i,k} = \frac{1}{N} \sum_{n=1}^N \|\mathcal{F}(f_i^k(R_n) - (f_i^k(F_n)) - M_n\|_2^2 \tag{9}$$

$\mathcal{F}$ represents normalization and scaling operations, $f_i^k \in \mathbb{R}^{W_i * H_i}, k \in [0, C_i]$, represents the feature map of the $k-$th dimension of the $f_i$. During the training process, we only select the channels reserved for $f_i$ during the AFFS initialization process. After AFFS processing, the original features $f_i \in \mathbb{R}^{W_i * H_i * C_i}$ will become $f_i' \in \mathbb{R}^{W_i * H_i * m_i}$.

## E.3 Visual Explanation of AFFS

Here we provide a set of visual examples of AFFS in Figure 13. The experiment was conducted on the 2-th layer of EfficientNetB4. Unlike the actual situation, in order to enhance readability, we only calculated $L_{AFFS}$ for a single epoch on a single pair of samples. And based on this, sort the 32 dimensions of the features in the 2-th layer. Here, we present the dimensions ranked in the top 14 and bottom 4. In the actual process, we iterate for 5 epochs on all samples on the training set to initialize AFFS. In Table 7, we present the changes in feature dimensions of different backbone features before and after performing AFFS.

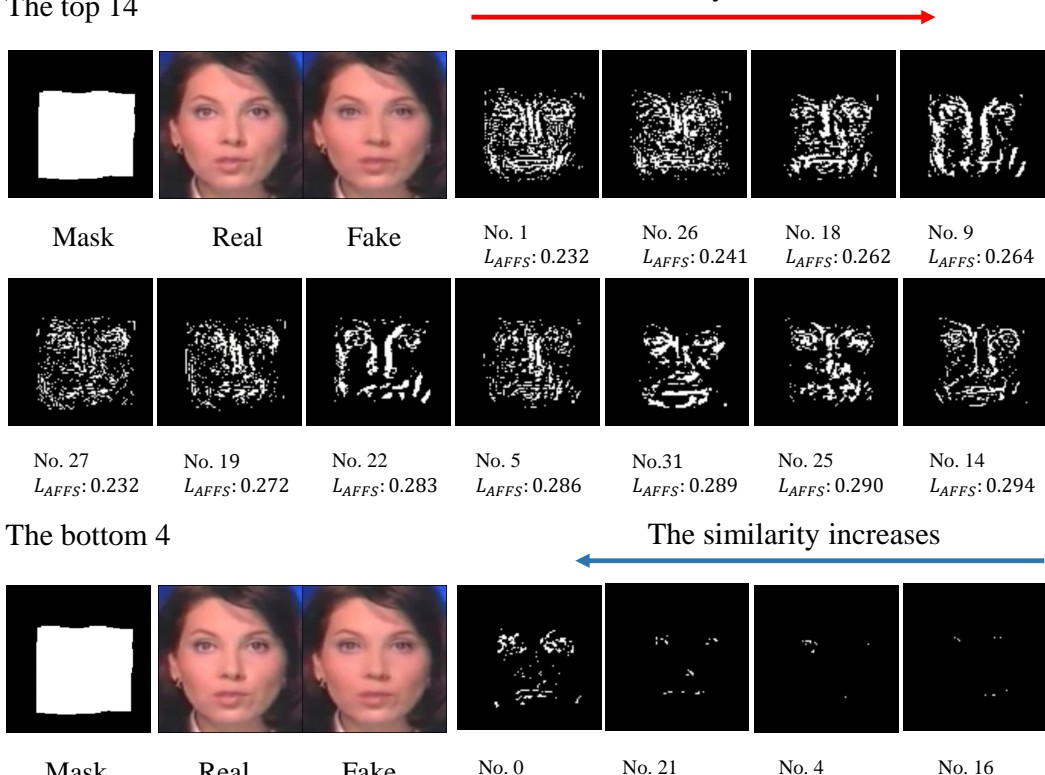

Figure 13: The result of calculating $L_{AFFS}$ for the 32 dimensions of the 2-th layer features of EfficientNetB4 on a single sample and sorting them by similarity.

Table 7: Feature dimension parameters of AFFS.

| Model | Input Dimension | Output Dimension |
|---|---|---|
| Res50 | {256,512,1024,2048} | {196,384,768,1536} |
| Res101 | {256,512,1024,2048} | {196,384,768,1536} |
| Effb1 | {24,40,112,320} | {16,24,40,112} |
| Effb4 | {32,56,160,448} | {24,32,56,160} |

Table 8: We report the Video-level AUC. The combination of our data augmentation method with other SOTA detection algorithms. † represents the method used in the original paper.

| Method | Training set | CDFv2 | DFDCP | Avg. |
|---|---|---|---|---|
| LAA-Net | SBI + EPPN † | 0.840 | 0.741 | 0.791 |
| LAA-Net | FIA-USA + EFPN | 0.875 | 0.782 | 0.829 |
| Ours | FIA-USA + FPN (AFFS) | 0.901 | 0.861 | 0.881 |
| Ours | FIA-USA + FPN (AFFS) +RCR | 0.941 | 0.866 | **0.904** |

## F    Details of RCR

In the LAA Net [51], a very interesting hypothesis is the pixels on the fake boundary contain both the features of the fake area and the features of the real area, so in our designed RCR, such pixels will be discarded. Figure 12 provides a visual example.

## G    Testing on more forgery techniques

In addition to several widely used forgery detection datasets mentioned in the paper, we also conducted experiments on other forgery techniques based on the DF40 dataset [80], where real samples were obtained from FF++ and fake samples were constructed using corresponding forgery techniques based on FF++ settings. As shown in the Table 9a, our proposed forgery detection framework has achieved excellent detection results on the vast majority of forgery methods. Meanwhile, Table 9b shows the comparison between our method and other state-of-the-art methods on various forgery techniques in DIFF.

Table 9: Test results on more forgery techniques.

(a) **We tested all forgery techniques provided on the DF40 dataset and reported Frame-Level AUC.** All results with AUC greater than 80 are highlighted in bold, while those with AUC less than 80 but greater than 70 are underlined.

| Type | mobileswap | MidJourney | faceswap | styleclip | DiT | lia | ddim | mcnet | RDDM | StyleGAN3 |
|---|---|---|---|---|---|---|---|---|---|---|
| AUC | **0.97** | **0.97** | **0.89** | **0.82** | 0.66 | **0.91** | **0.97** | **0.84** | 0.70 | **0.93** |
| Type | e4e | pixart | whichfaceisreal | deepfacelab | StyleGANXL | heygen | facedancer | MRAA | pirender | VQGAN |
| AUC | **0.95** | **0.93** | 0.42 | 0.77 | 0.41 | 0.58 | **0.83** | **0.85** | **0.81** | **0.85** |
| Type | StyleGAN2 | facevid2vid | simswap | inswap | one_shot_free | wav2lip | e4s | starganv2 | tpsm | sd2.1 |
| AUC | **0.93** | **0.83** | **0.91** | **0.87** | **0.85** | 0.76 | **0.88** | 0.48 | **0.83** | **0.97** |
| Type | blendface | hyperreenact | uniface | CollabDiff | fsgan | danet | SiT | sadtalker | fomm | |
| AUC | **0.94** | **0.80** | **0.92** | **0.95** | **0.86** | **0.80** | 0.72 | 0.74 | **0.85** | |

(b) **Comparison with universal deepfake detection methods using the Frame-Level AUC on the DiFF dataset.** The notation † indicates models that are designed for deepfake detection, while ‡ signifies models intended for general generated detection. All experiments were trained on the c23 version of FF++. The best result is bolded.

| Method | Test Subset | | | | | | | | | | | | |
|---|---|---|---|---|---|---|---|---|---|---|---|---|---|
| | Cycle | CoDiff | Imagic | DiFace | DCFace | Dream | SDXL_R | FD_I | LoRA | Midj | FD_T | SDXL | HPS |
| F$^3$-Net† [56] | 36.14 | 35.00 | 32.56 | 25.61 | 53.29 | 55.45 | 65.04 | 40.90 | - | - | 45.00 | 61.64 | 68.96 |
| EfficientNet† [66] | 56.51 | 38.38 | 48.50 | 64.45 | 89.13 | 71.64 | 65.04 | 59.93 | - | - | 69.67 | 64.94 | 74.63 |
| CNN_Aug‡ [70] | 50.31 | 46.84 | 75.95 | 43.10 | 80.69 | 58.75 | 60.10 | 43.65 | - | - | 47.90 | 61.95 | 60.56 |
| SBI† [63] | 82.32 | 64.70 | 49.84 | 73.64 | 89.92 | 75.66 | 77.73 | 87.67 | 89.18 | 79.56 | 90.55 | 80.37 | 83.30 |
| Ours | **84.76** | **72.61** | 52.92 | **76.86** | **93.63** | **83.43** | **80.54** | **92.68** | **93.48** | **82.57** | **94.85** | **82.15** | **88.40** |

## H    Combining with SoTA detection method

In this section, we aim to validate the effectiveness of our FIA-USA data augmentation by integrating it with state-of-the-art image-level detection methods. While our comprehensive evaluation currently focuses on LAA-Net [51] (CVPR 24) due to limited availability of open-source image-level detection implementations, Table 8 reveals significant performance improvements. **Notably, the original LAA-Net implementation was trained on raw version FF++ data, whereas our experiments utilize the c23 version.** Compared with the baseline implementation, our data augmentation strategy has significantly improved the performance of the original detector. To address requests for detailed comparisons with LAA-Net, our table shows two key findings: First, our method achieves a 7.5% performance advantage over LAA-Net when using identical data augmentation conditions. Second, even without employing RCR (Robust Context Refinement), our approach substantially outperforms LAA-Net's EFPN component, demonstrating the inherent strength of our core methodology.

## I    Limitations

The proposed detection framework primarily focuses on **image-level data augmentation** and matching detection architectures. While we categorize image-level artifacts into FIA and USA, and have achieved satisfactory video-level detection performance, we acknowledge that video-level artifacts present more complex challenges - including temporal inconsistencies and audio-visual asynchrony - which fall outside the current research scope but represent our intended future research direction.

Second, although recent approaches increasingly incorporate large models' zero-shot capabilities for deepfake

detection, existing work has not yet explored the integration of data augmentation methods with such models due to practical constraints including variations in training methodologies and model scales. This does not mean that our method cannot be applied to large models, but currently we are still unfamiliar with large models and have not fully explored this part.

Regarding experimental validation, **we mainly compared and analyzed our proposed method with previous data augmentation methods, which is consistent with the main contribution of our paper**. Of course, we also compared state-of-the-art detection methods that are not data augmentation based. Following established practices in data augmentation research, we conducted comprehensive evaluations across four widely adopted network architectures to ensure fair comparison. However, we note that detection performance may vary across different network frameworks depending on their baseline performance and inherent compatibility with deepfake detection tasks. We cannot guarantee that all model frameworks will achieve excellent detection results after using our method.

Finally, while our method demonstrates effectiveness across a substantial majority of forgery methods (**validated on over 58 different techniques**), we acknowledge relatively weaker performance in specific edge cases. Nevertheless, the framework maintains robust detection capability for most common forgery approaches, achieving satisfactory overall performance that meets our research objectives.

## J Broader impacts

The proposed framework offers positive societal impacts by enhancing detection of sophisticated face deepfakes, thereby mitigating disinformation campaigns, fake news, and fraudulent content that erode public trust in digital media. Additionally, it safeguards individual privacy by identifying forged facial manipulations that enable identity theft, unauthorized face swapping, and other privacy violations. The method could further support content authenticity initiatives through integration into social media platforms or news verification systems to prioritize legitimate visual content. However, potential negative impacts include the risk of adversaries reverse-engineering the detection framework to refine deepfake generation methods, potentially escalating the adversarial "arms race" between forgery creators and detectors. Furthermore, false positives in detection could lead to unintended censorship, where legitimate content is erroneously flagged as fake.

## K More visual examples

From Figure 14 to Figure 16, we demonstrate the fake samples that FIA-USA can generate through different reconstruction methods, including autoencoderr (AE) and facial super-resolution (SR), and multiple types of masks based on three real faces. Figure 17 shows the negative samples randomly generated by FIA-USA during the real training process. Figures 18 to 22 list the images in the DF40 dataset [80] that were misclassified by our method.

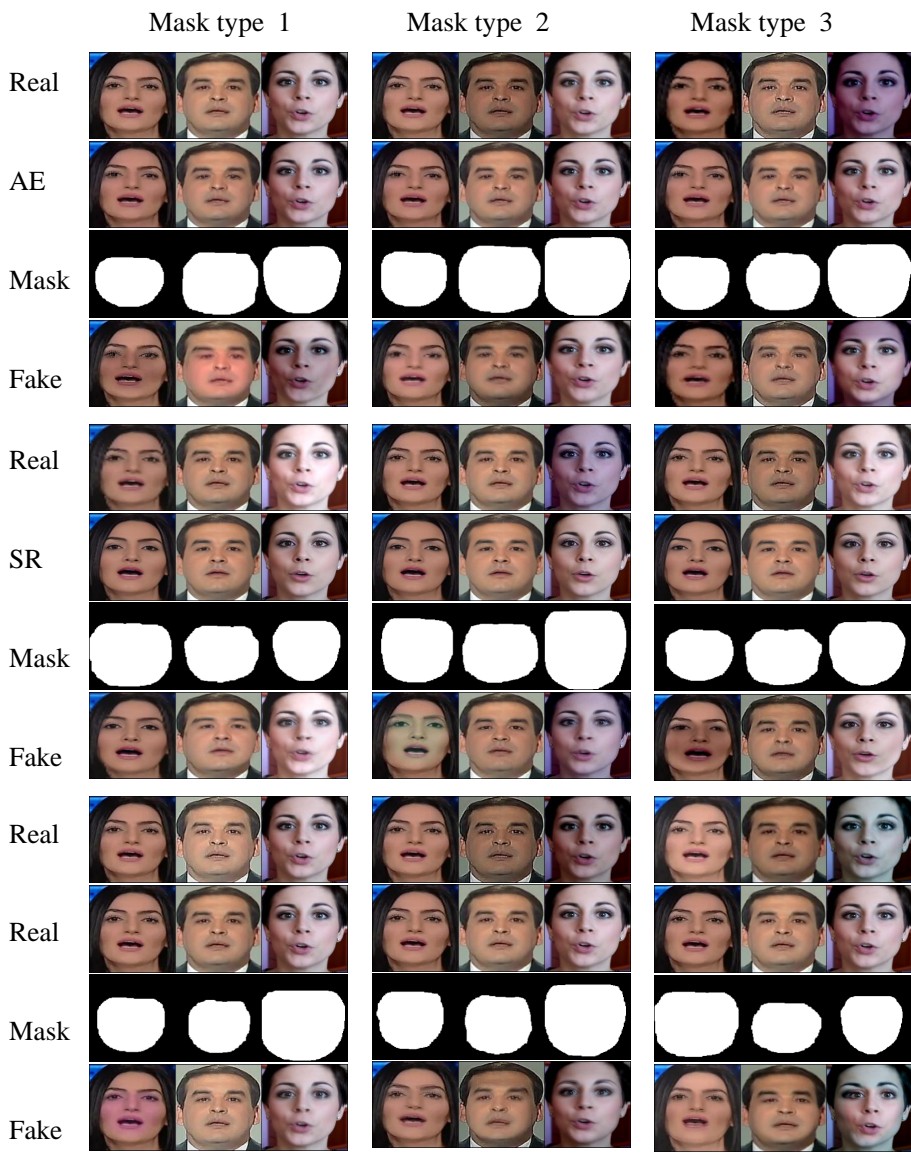

Figure 14: **More examples of FIA-USA.**

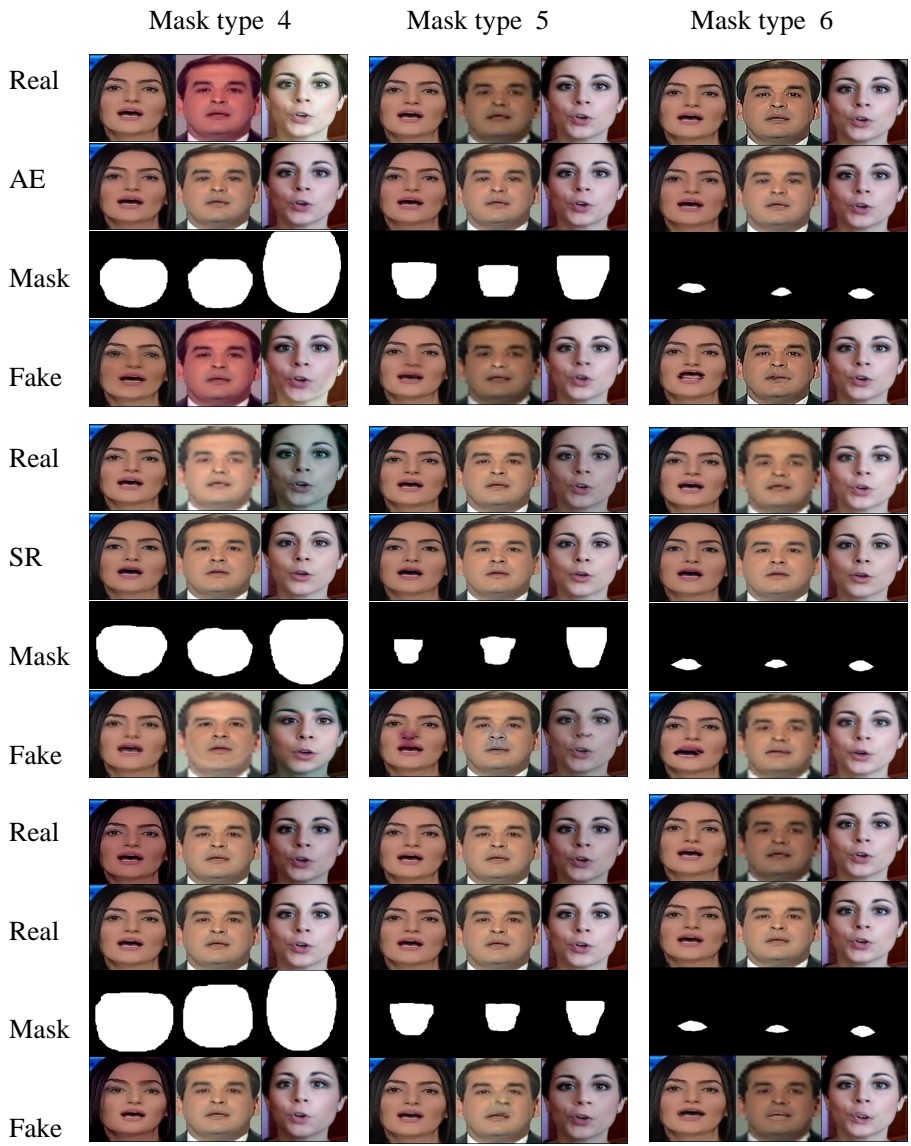

Figure 15: **More examples of FIA-USA.**

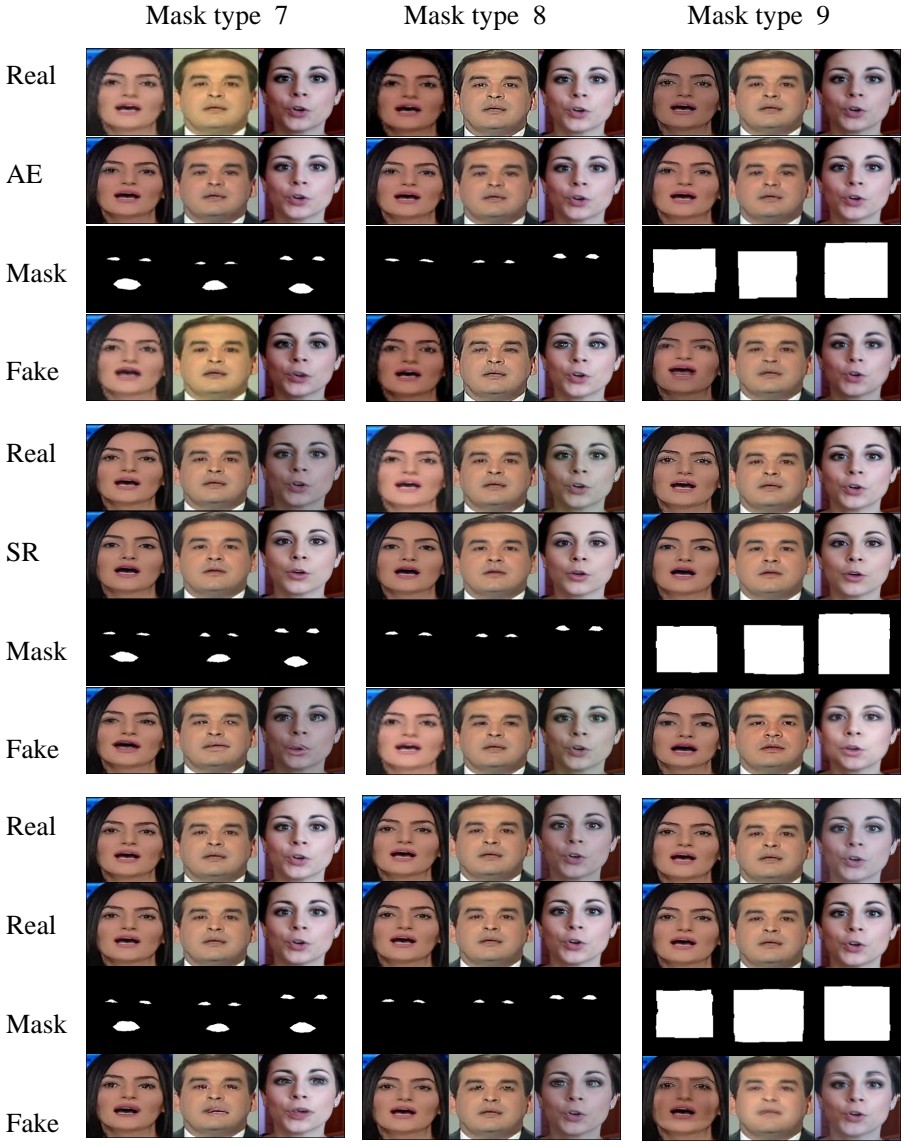

Figure 16: **More examples of FIA-USA.**

Real

Reconstructed

Mask

Fake

Real

Reconstructed

Mask

Fake

Real

Reconstructed

Mask

Fake

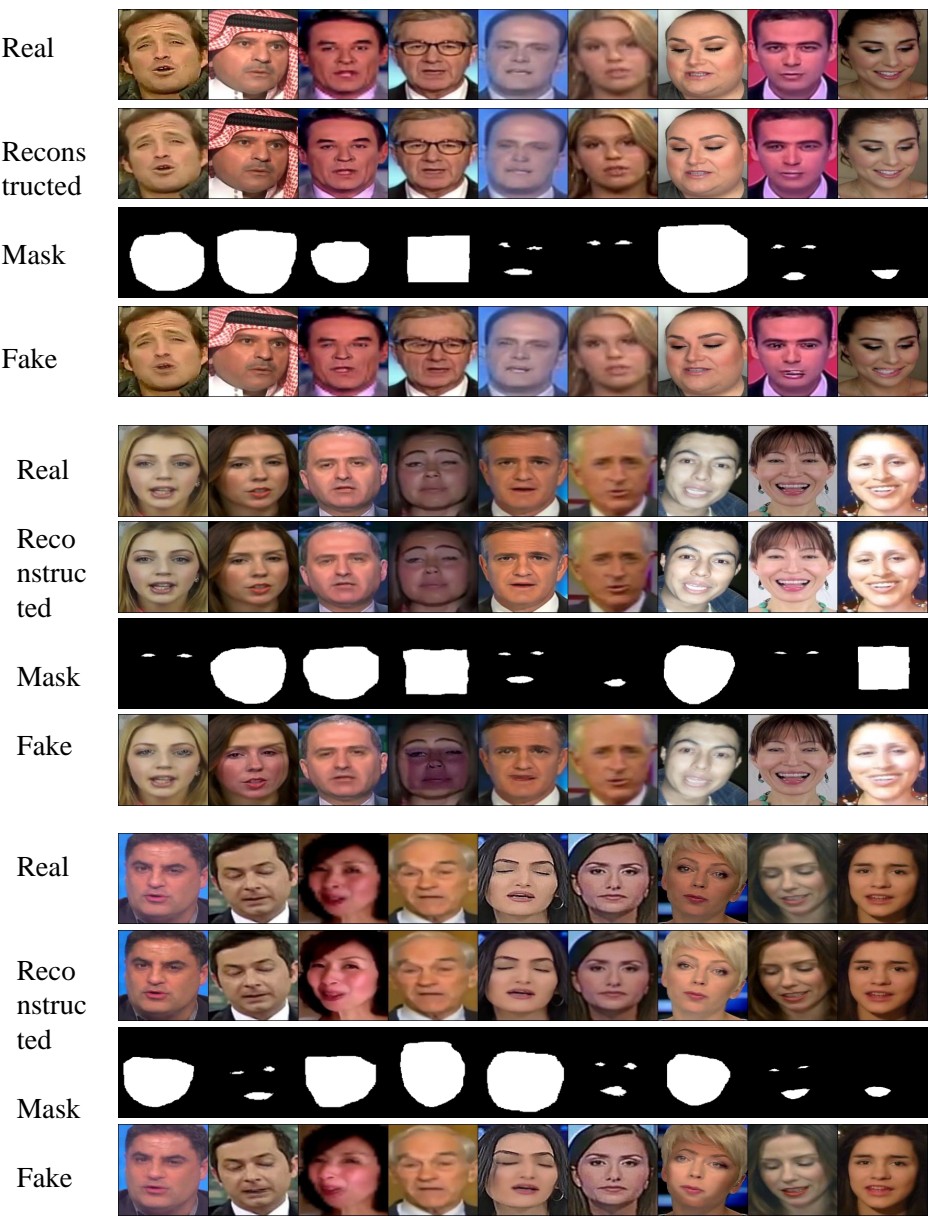

Figure 17: **More examples of FIA-USA.**

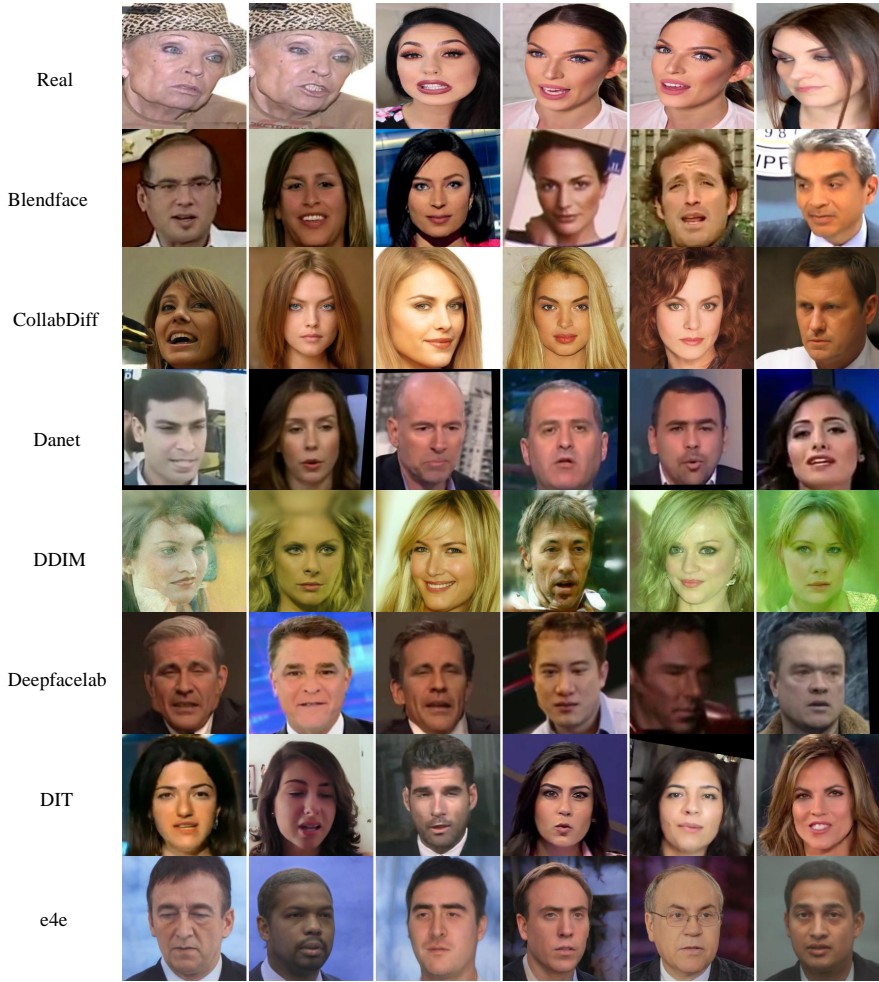

Figure 18: **Images misclassified by our method in the DF40 dataset.**

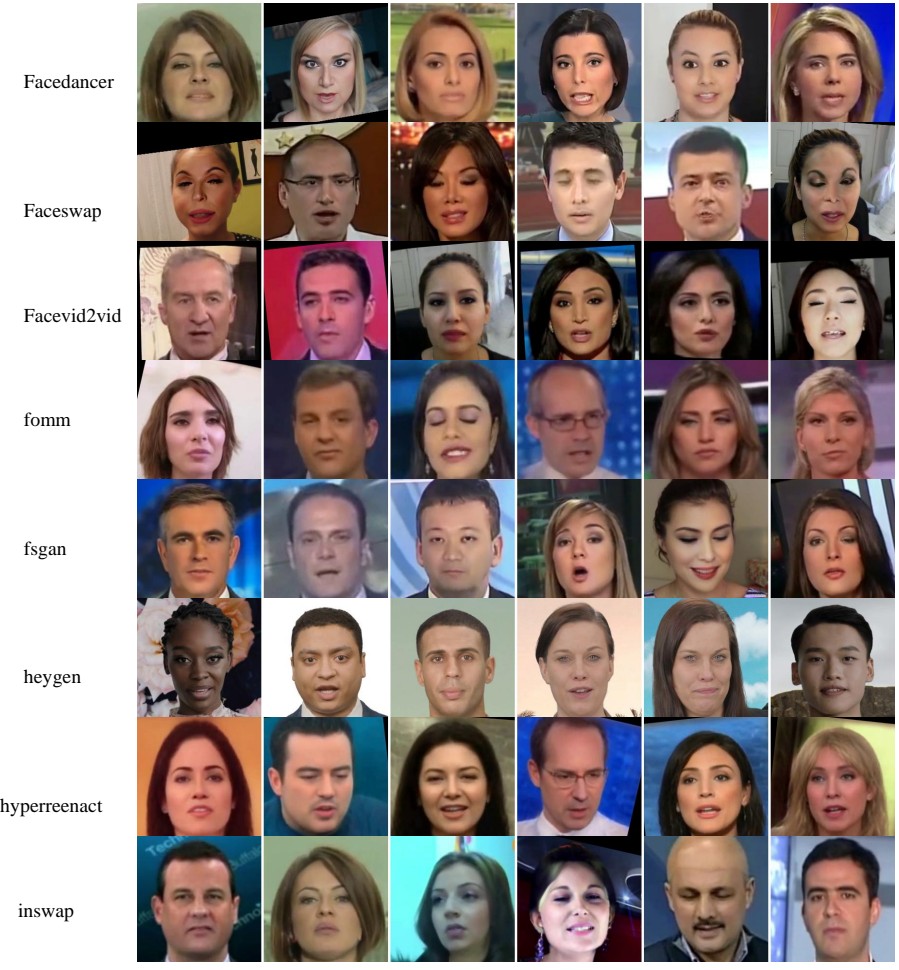

Figure 19: **Images misclassified by our method in the DF40 dataset.**

mcnet

MidJourney

mobileswap

MRAA

one_shot_free

pirender

pixart

RDDM

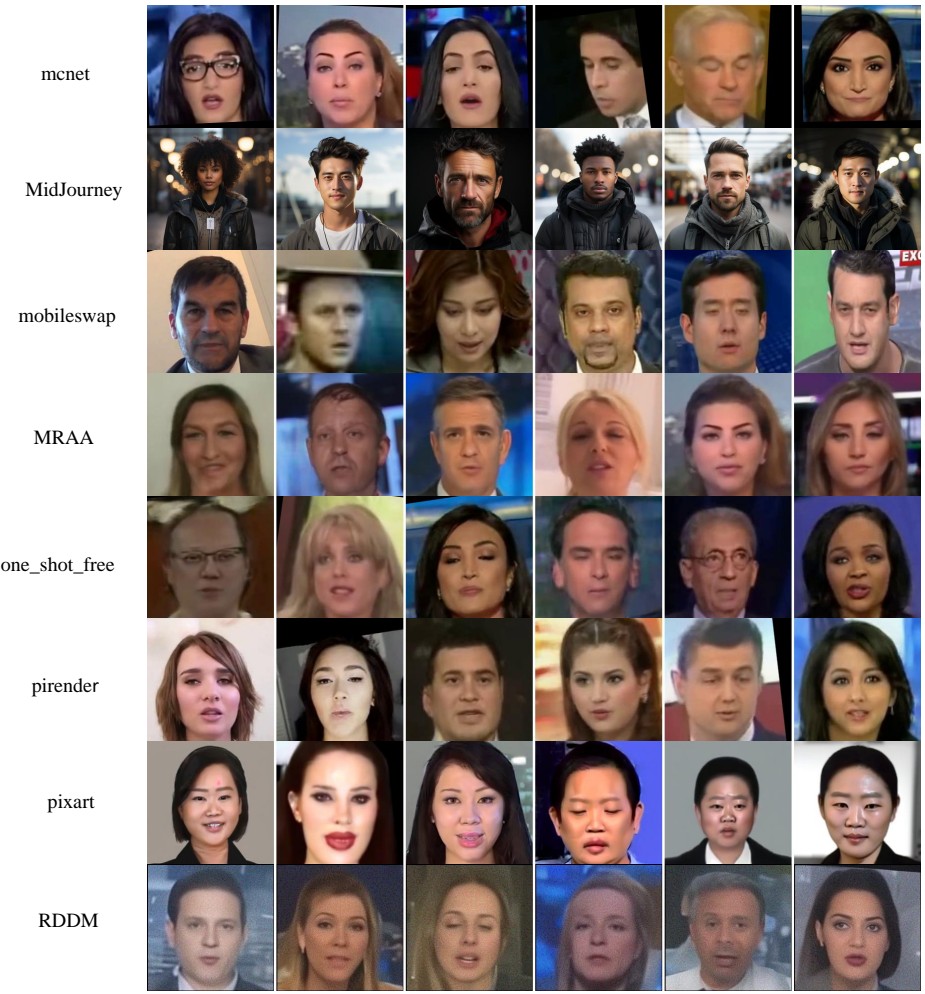

Figure 20: **Images misclassified by our method in the DF40 dataset.**

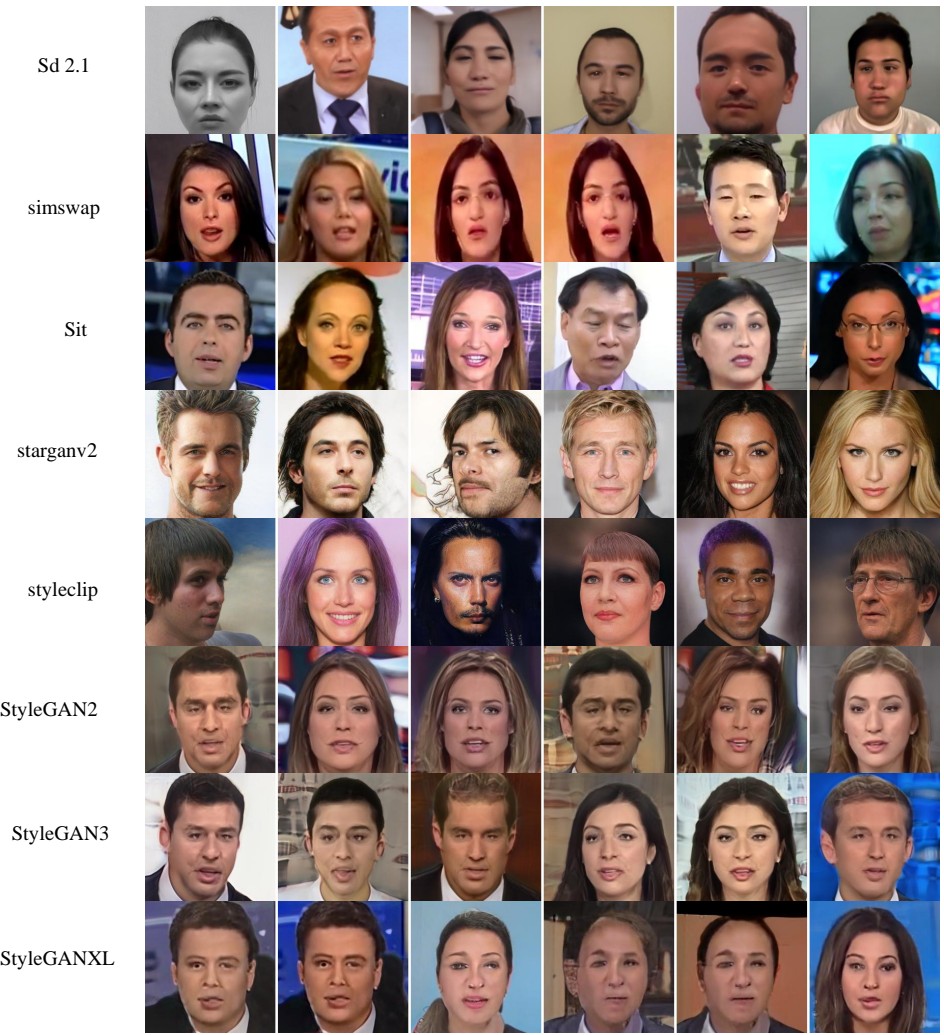

Figure 21: **Images misclassified by our method in the DF40 dataset.**

uniface

VQGAN

Wav2lip

whichfaceisreal

e4s

lia

tpsm

Sadtalker

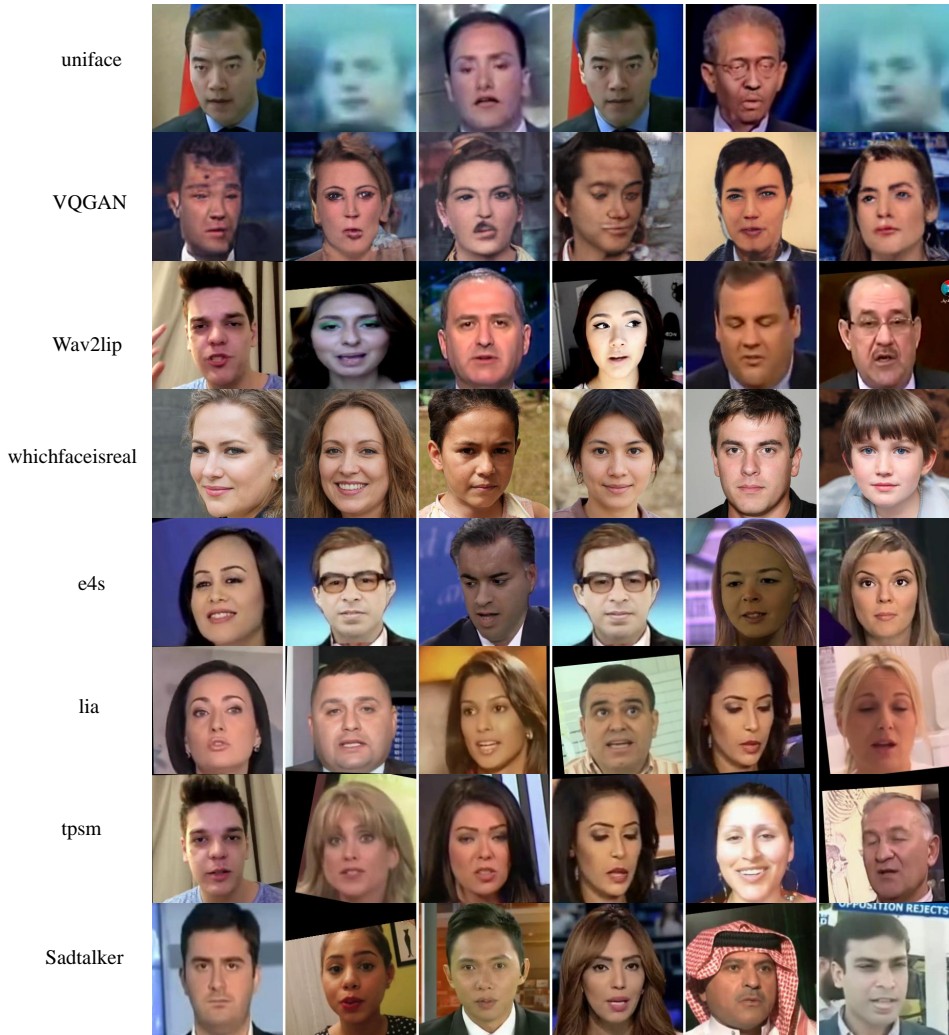

Figure 22: **Images misclassified by our method in the DF40 dataset.**

