# OpenReview forum: "From Specificity to Generality: Revisiting Generalizable Artifacts in Detecting Face Deepfakes"
_NeurIPS.cc/2025/Conference — NeurIPS 2025 poster_

### Official Review · Reviewer_uxGL · 2025-06-22

**Clarity:** 2
**Significance:** 3
**Originality:** 3
**Rating:** 5
**Confidence:** 5

**Summary:**

Briefly summarize the paper and its contributions. This is not the place to critique the paper; the authors should generally agree with a well-written summary. This is also not the place to paste the abstract—please provide the summary in your own understanding after reading.

This paper introduces a universal framework for detecting facial deepfakes by focusing on identifying common and generalizable forgery artifacts. The core challenge addressed is the wide diversity of existing deepfake generators and the impracticality of teaching a detector to learn all varied artifact types separately.

The authors propose categorizing deepfake artifacts into two distinct yet complementary types:

Face Inconsistency Artifacts (FIA): These arise from the difficulty of generating intricate facial details, leading to inconsistencies between complex facial features and their surrounding areas, or from blending boundaries in face swapping.

Up-Sampling Artifacts (USA): These are inevitable traces left by the generator’s decoder during the up-sampling process, as decoding and up-sampling are fundamental operations in facial manipulation generators. The authors claim that all existing deepfakes are observed to exhibit either or both of these artifact types.

To enable detection models to learn these generalizable artifacts, the paper proposes an image-level pseudo-fake creation framework called FIA-USA. This framework constructs fake samples by simulating both FIA and USA without introducing less-general artifacts. Specifically, it reconstructs target faces to simulate USA using autoencoders (AE) or super-resolution (SR) models, and utilizes image-level blending with multiple mask types on diverse facial regions to create FIA. The blending process includes macro-editing masks (based on 4 or 81 facial keypoints) and micro-editing masks (randomly combining facial features like eyes and mouth).

Beyond the data augmentation, the framework also includes two complementary techniques to maximize efficacy:

Automatic Forgery-aware Feature Selection (AFFS): This streamlines feature dimensionality and augments feature discernment by adaptively selecting channels in Feature Pyramid Networks (FPN) that are sensitive to forgery regions, thereby reducing feature redundancy.

Region-aware Contrastive Regularization (RCR): By juxtaposing features of manipulated regions against authentic regions, RCR enhances the model's focus on upsampling traces and inconsistencies between different areas. This includes both Patch-level Contrastive Regularization (PCR) and Image-level Contrastive Regularization (ICR).
Extensive experiments on seven widely used deepfake detection datasets, encompassing over 58 distinct forgery methods, validate the superior efficacy and robustness of the proposed method. The results demonstrate that a standard image classifier trained with FIA-USA pseudo-fake data can generalize well to unseen deepfakes, even outperforming state-of-the-art methods, including those based on latent space augmentation. The paper concludes that this targeted approach, by focusing on fundamental artifact patterns, can help address challenges in current detection methods and provide a promising foundation for more adaptable deepfake detection systems.

**Questions:**

Question: The paper states that its core methodology for detecting deepfakes primarily relies on analyzing image-level artifacts, namely Face Inconsistency Artifacts (FIA) and Up-Sampling Artifacts (USA). While the evaluation includes "satisfactory video-level detection performance", the framework acknowledges that "video-level artifacts present more complex challenges - including temporal inconsistencies and audio-visual asynchrony - which fall outside the current research scope". How does the current image-level focused approach implicitly capture or respond to these video-level inconsistencies?  Is this a future area of research interest?

Proposed Suggestion: For the authors to strengthen the paper's contribution towards a "universal detection framework", it would be beneficial to elaborate on the specific temporal or inter-frame cues (if any) that their current image-based approach implicitly leverages to achieve reported video-level AUC scores. If applicable, could the authors provide a brief discussion or hypothesis on how the FIA-USA framework might be extended or complemented in future work to directly address these complex video-level artifacts (e.g., inconsistencies over time or audio-visual discrepancies), rather than solely relying on aggregating frame-level detections? This would clarify the path towards a truly universal solution for all types of deepfakes, including video-specific forgeries.

Question: The paper explicitly acknowledges that it has "not yet explored the integration of data augmentation methods with such [large] models due to practical constraints". This implies a current boundary to the method's general applicability, especially as larger models increasingly dominate deepfake generation and detection.

Proposed Suggestion: While a full exploration of large model integration is noted as future work, could the authors expand on the primary challenges they anticipate when adapting or integrating their FIA-USA framework, AFFS, and RCR with large foundation models (e.g., Vision Transformers, very large Diffusion Models)? Are these challenges primarily computational, architectural, or related to the nature of artifacts produced by and detectable by these larger models? Providing this insight would enhance the paper's forward-looking perspective and set a clearer agenda for future research in this area.

Question: The paper states that all experiments were conducted on "a single NVIDIA 3090" and indicates that resource requirements are "minimal". However, specific metrics like execution time for training/inference or precise memory consumption are not provided in the main text or the appendix beyond this general statement.

Proposed Suggestion: To ensure full experimental reproducibility and provide practical insights for researchers and practitioners considering deploying this method, could the authors provide approximate training times (e.g., per epoch or total) and inference times (e.g., frames per second) for their model on a standard dataset (e.g., FF++) using the specified hardware? Additionally, GPU memory consumption (e.g., VRAM usage) would be a useful detail. This specific data would allow others to more accurately gauge the computational feasibility and scalability of the proposed approach.

**Ethical Concerns:**

["NO or VERY MINOR ethics concerns only"]

**Final Justification:**

I appreciate the thorough rebuttal by the authors and after careful consideration I have decided to raise my score to accept.

**Limitations:**

Yes.

**Paper Formatting Concerns:**

None.

**Quality:**

3

**Strengths And Weaknesses:**

Quality

Strengths:
* Robust Experimental Validation: The paper conducts extensive experiments on seven widely used Deepfake detection datasets, encompassing over 58 distinct forgery methods, including identity swapping, expression reenactment, generative face synthesis, and attribute editing. This broad evaluation validates the superior efficacy and robustness of their method.

*  Performance: The proposed method is shown to outperform other state-of-the-art (SOTA) approaches on the majority of traditional deepfake datasets and achieves competitive results on others, such as CDF-v1. Notably, it demonstrates strong generalization capabilities on generative deepfake datasets like DiFF and DF40, often surpassing previous methods by significant margins (e.g., 9.7% over RECCE and 23.4% over SBI on DF40 average AUC).

*  Robustness: The method exhibits superior robustness to unseen perturbations like Gaussian Blur, Block Wise, Change Contrast, and JPEG Compression, as quantified by the reduction rate of Video-Level AUC.

* Ablation Studies: Detailed ablation experiments confirm the effectiveness of each component (FIA-USA, AFFS, RCR), showing a progressive degradation in performance when individual components are removed, which strengthens the argument for their contributions.

* Technical Soundness: The categorization of artifacts (FIA and USA) is grounded in a systematic analysis of shared technical frameworks in existing deepfake algorithms. The methodology for constructing pseudo-fakes (FIA-USA) is well-articulated, involving multi-type mask synthesis and multi-modal reconstruction using autoencoders and super-resolution models to simulate realistic artifacts. The integration of AFFS and RCR further refines feature learning for robust detection.

Weaknesses:

* Limited Evaluation of Full Video-Level Artifacts: While the paper achieves satisfactory video-level detection performance and acknowledges video-level artifacts as a future direction, the primary focus remains on image-level data augmentation and detection architectures. This leaves complex video-level challenges like temporal inconsistencies and audio-visual asynchrony largely unexplored in the current framework .

* Unexplored Large Model Integration: The authors note that they have not yet explored the integration of their data augmentation methods with large models' zero-shot capabilities, citing practical constraints and unfamiliarity with large models. This could be a limitation in light of recent advancements in large generative models.

* Performance in Edge Cases: Although the method is effective across a substantial majority of forgery methods (over 58), the authors acknowledge relatively weaker performance in specific edge cases. While they state it maintains robust detection for most common approaches, these edge cases could be critical.

Clarity


Strengths:

* Detailed Methodology: The core components, especially FIA-USA, AFFS, and RCR, are explained in detail, including their motivations and mathematical formulations. The appendix further elaborates on mask generation, reconstruction models, and the theoretical basis of AFFS.

* Sufficient Experimental Details: The paper provides comprehensive details for reproducibility, including the backbone network architecture, optimizer, batch size, learning rate, training epochs, video sampling, data augmentation pipeline, loss coefficients, and hardware used (NVIDIA 3090). Data splits and datasets are clearly specified, including how FF++ was used for training.

Weaknesses:

* Lack of Open-Source Code: The authors explicitly state that the code is "currently not open source". While the paper provides extensive details, the absence of publicly available code can hinder direct reproducibility and further research by the community, despite the detailed instructions. This is noted as a "No" in the NeurIPS checklist .

* Limited Compute Resource Details: While the GPU type is specified (NVIDIA 3090), more granular details such as memory usage and exact execution times for experiments are not provided. While authors claim this is unlikely to affect reproducibility for experiments of this scale, it could be beneficial for researchers with different computational resources.

Significance

Strengths:
*  The paper tackles the increasingly important topic of deepfake detection, driven by the rapid development of AI generation techniques and the associated risks of privacy violations and misinformation spread. Developing a universal deepfake detector is highlighted as an "urgent priority".

* Generalizable Approach: By focusing on common and generalizable artifacts (FIA and USA) instead of specific ones, the framework moves towards a universal detection capability, enhancing the generalization of detection models across diverse and unseen deepfake variations. This is a significant step beyond methods tailored for specific forgery techniques.

Originality

Strengths

* Novel Pseudo-Fake Creation Framework: The paper introduces a "novel image-level pseudo-fake creation framework" (FIA-USA) which is a significant original contribution. Unlike prior methods that often focus on isolated artifact types or single-scale modifications, FIA-USA is a dual-artifact collaborative enhancement framework designed for universal detection, addressing both global/local and facial/regional alterations, and specifically introducing USA, which was less emphasized previously.

* Complementary Techniques: The introduction of Automatic Forgery-aware Feature Selection (AFFS) and Region-aware Contrastive Regularization (RCR) as complementary techniques further contributes to the originality. AFFS is described as a "statistically-driven feature compression paradigm" that quantifies channel-wise sensitivity to forgery regions, and RCR explicitly models divergence between forged and authentic regions through multi-granularity contrastive learning.

Weaknesses:
* Building on Existing Blend Paradigm: While FIA-USA injects new vitality, it builds upon an existing blend-based data augmentation paradigm. The originality lies in how they extend this paradigm (multiple masks, USA introduction) rather than inventing blending from scratch. This is a minor point, as the authors clearly explain their advancements over previous blending methods.

---

> ### Author Rebuttal · Authors · 2025-07-31
>
> Dear reviewer uxGL:
>
> On behalf of all authors, we sincerely thank you for your thorough evaluation and generous recognition of our work. We are deeply encouraged by your positive remarks on the **rigorous experimental design, robustness of results, exceptional performance benchmarks, and meticulous ablation studies** – your insightful feedback is invaluable to our research development. Below, we provide point-by-point responses to your specific questions to further clarify and enhance the manuscript.
>
> > **Q1.**  Could the authors briefly discuss how the FIA-USA framework might be extended or enhanced to directly address complex video-level artifacts in future work, rather than relying solely on aggregating frame-level detections?
>
> **R1.** Thank you for recognizing our work. For video-level detection, our current approach is to uniformly sample 32 frames from a video and compute the average forgery probability across these frames to represent the video-level probability.
>
> Prior to this work, we try to develop a series of video data augmentation methods. These methods employ frame-level alterations (e.g., introducing artifacts using SBI) and induce temporal inconsistencies by frame rearrangement. However, this method does not significantly enhance the performance of model. Our current paper primarily explores image-level data augmentation, we find that robust detectors can be trained effectively using image-level augmentation alone. This finding suggests that developing meaningful video-level data augmentation likely requires strategies to generate more generalizable temporal artifacts. Indeed, overly simplistic temporal inconsistencies can actually degrade model performance.
>
> **A potentially promising direction is to incorporate acoustic features to create such inconsistencies, or integrate physiological signals (e.g., eye blinking patterns, facial movement patterns) into the augmentation pipeline.** However, we must honestly report that despite sustained exploration in this direction, we have yet to achieve significant breakthroughs.
>
>
> ---
>
> > **Q2.**  Performance in Edge Cases: Although the method is effective across a substantial majority of forgery methods (over 58), the authors acknowledge relatively weaker performance in specific edge cases.
>
>
> **R2.**  Regarding model performance, our method currently yields only 5 unsatisfactory detection results when evaluated against over 58 distinct forgery methods, demonstrating considerable effectiveness. Nevertheless, we intend to pursue further improvements.
>
> **Table 9:**  **We test all forgery techniques provided on the DF40 dataset and reported Frame-Level AUC.** All results with AUC greater than 80 are **highlighted** in bold, while those with AUC less than 80 but greater than 70 are **underlined**.
>
>  **Type** | mobileswap    | MidJourney    | faceswap        | styleclip        | DiT             | lia               | ddim             | mcnet            | RDDM             | StyleGAN3
> :--------:|:-------------:|:-------------:|:---------------:|:----------------:|:---------------:|:-----------------:|:----------------:|:----------------:|:----------------:|:-----------------------------:
>  **AUC**  | $\textbf{0.97}$ | $\textbf{0.97}$ |$\textbf{0.89}$   | $\textbf{0.82}$   | 0.66            | $\textbf{0.91}$     | $\textbf{0.97}$    | $\textbf{0.84}$    | $\underline{0.70}$ | $\textbf{0.93}$
>  **Type** | e4e           | pixart        | whichfaceisreal | deepfacelab      | StyleGANXL      | heygen            | facedancer       | MRAA             | pirender         | VQGAN
>  **AUC**  | $\textbf{0.95}$ | $\textbf{0.93}$ | 0.42            | $\underline{0.77}$ | 0.41            | 0.58              | $\textbf{0.83}$    | $\textbf{0.85}$    | $\textbf{0.81}$    | $\textbf{0.85}$
>  **Type** | StyleGAN2     | facevid2vid   | simswap         | inswap           | one\_shot\_free | wav2lip           | e4s              | starganv2        | tpsm             | sd2.1
>  **AUC**  | $\textbf{0.93}$ | $\textbf{0.83}$ | $\textbf{0.91}$   | $\textbf{0.87}$    | $\textbf{0.85}$   | $\underline{0.76 }$ | $\textbf{0.88}$    | 0.48             | $\textbf{0.83}$    | $\textbf{0.97}$
>  **Type** | blendface     | hyperreenact  | uniface         | CollabDiff       | fsgan           | danet             | SiT              | sadtalker        | fomm             |
>  **AUC**  | $\textbf{0.94}$ | $\textbf{0.80}$ | $\textbf{0.92}$   | $\textbf{0.95}$    | $\textbf{0.86}$   | $\textbf{0.80}$     | $\underline{0.72}$ | $\underline{0.74}$ | $\textbf{0.85}$    |
>
> ---
>
> > **Q3.**  What are the main challenges (computational, architectural, or artifact-related) in integrating FIA-USA/AFFS/RCR with large foundation models? This would clarify future research directions.
>
> **R3.** The primary challenge with VLM (Vision-Language Model) integration stems from our team's limited prior exposure to VLMs. We remain unfamiliar with their training paradigms, fine-tuning strategies, and underlying architectures. Although our FIA-USA framework can generate samples suitable for VLM training, we suspect simple feature concatenation may yield insufficient gains. Our future direction involves actively exploring VLMs to develop more robust detectors. However, this will require substantial research time.
>
> Critically, we believe **VLM's core superiority lies in its interactive capabilities, intrinsic interpretability, and task universality** – attributes that must be fully leveraged. Without adapting to these strengths, VLM adoption risks becoming merely a transition from one encoder architecture to another, offering marginal benefit. **Consequently, extending FIA-USA to VLMs will not treat them as mere classifiers. Instead, we aim to propose either: A more interpretable framework, or A unified multi-task framework (e.g., incorporating forgery localization).** Achieving this will likely require advanced large-model training techniques, such as reinforcement learning-based fine-tuning approaches.
>
> ---
> > **Q4.** Lack of Open-Source Code.
>
> **R4.** We strongly support open source code. However, as our paper encompasses extensive experimentation, multiple methodologies, and numerous interdependent components, organizing the codebase presents significant challenges. Releasing hastily structured code would likely compromise readability and usability and may introduce technical issues, potentially consuming reviewers' valuable time unnecessarily. Please rest assured that we are currently undertaking a thorough codebase restructuring to ensure high quality and usability. **We plan to open-source the code promptly upon the paper's acceptance.**
>
> ---
>
> > **Q5.**  Provide the resources needed for reproduction：1. Training time(per epoch/total).  2. Inference speed(frames per second). 3. GPU memory.
>
> **R5.** For model training and inference, we utilize the FF++ C23 version for real faces. The training set comprise 720 real videos, from which we extracted 32 frames per video, resulting in a total of 720 * 32 * 2 training images (720 * 32 real faces + 720 * 32 FIA-USA generated fake faces ). Each training epoch take approximately 48 minutes, requiring 20.2 GB of GPU memory with a batch size of 12. The validation set consist of 140 real videos, yielding 140 * 32 * 2 images. Each validation epoch take about 3 minutes. We train the model for a total of 50 epochs, performing validation once per epoch. Before training commenced, we initialize the AFFS module, initializing parameters across 4 layers—a process taking approximately 20 minutes and requiring 1.7 GB of GPU memory. Consequently, the total expected training time is 42.8 hours. The speed of inference and validation is projected to be 50 images per second，requiring 20.2 GB of GPU memory with a batch size of 12.
>
>
>
>
> **Table R5: Resource Utilization**
> | Phase          | Duration              | GPU Memory | Batch Size | Frequency          | Details                       |
> |----------------|-----------------------|------------|------------|--------------------|-------------------------------|
> | Initialization | 20 minutes           | 1.7 GB     | -          | Once               | AFFS module setup (4 layers) |
> | Training       | 48 minutes/epoch     | 20.2 GB    | 12         | 50 epochs          | Per-epoch processing         |
> | Validation     | 3 minutes/run        | 20.2 GB    | 12         | 50 epochs  | Per-epoch evaluation         |
> | **Total Training** | **42.8 hours**   | -          | -          | -                  | Initialization + 50×(Train+Val) |
> | Inference      | 50 images/second     | 20.2 GB    | 12         | -                  | -    |

---

> ### Author Response · Authors · 2025-08-08
>
> Dear Reviewer uxGL,​​
>
> We sincerely thank you for your valuable feedback ​​and​​ recognition of our work. As the discussion phase is nearing its conclusion, we would like to confirm whether our previous responses have  addressed your concerns. Should any points remain unclear, we would be ​​pleased​​ to engage in further discussion.
>
> We truly appreciate your ​​feedback​​ and thank you once again for your time ​​and thoughtful assessment​​.
>
> Best regards,
>
> The Authors

---

> ### Comment · Reviewer_uxGL · 2025-08-08
> **Replying to author rebuttals**
>
> I appreciate the thorough rebuttal by the authors and after careful consideration I have decided to raise my score to accept.

---

### Official Review · Reviewer_FW6y · 2025-06-29

**Clarity:** 3
**Significance:** 3
**Originality:** 3
**Rating:** 5
**Confidence:** 4

**Summary:**

The paper proposes a universal deepfake detection framework that identifies and exploits two general forgery artifacts—FIA and USA—across various types of deepfake generation. The authors introduce a novel dual-artifact augmentation technique (FIA-USA) for pseudo-deepfake sample synthesis, along with two training enhancements: AFFS and RCR. Extensive experiments demonstrate great generalization and robustness compared to SOTA methods.

**Questions:**

1. Why you only choose (2.5, 1, 0.25) as your hyperparameters ($\lambda_1, \lambda_2, \lambda_3$)?
2. See Weaknesses.

**Ethical Concerns:**

["NO or VERY MINOR ethics concerns only"]

**Final Justification:**

I appreciate the clear rebuttal made by the authors. After considering the authors' response, I decided to maintain my original score.

**Limitations:**

Yes.

**Paper Formatting Concerns:**

None.

**Quality:**

4

**Strengths And Weaknesses:**

### Strengths
1. The FIA-USA data augmentation strategy is simple yet remarkably effective.
2. Extensive experiments on a wide range of datasets and rigorous ablation studies.
3. The method has good generalizability to unseen methods and robust under perturbations.

### Weaknesses
1. The brackets in Eq. 3 are not closed, making it unclear.
2. Although FIA-USA shows great effectiveness, it may heavily relies on the quality of pseudo-fake generation; performance might drop when faced with adversarial attacks or higher-fidelity forgeries.
3. The paper lacks a formal theoretical justification for the strong cross-domain generalization performance of FIA and USA. Providing such analysis would strengthen the claims.
4. There lacks a detailed analysis about the results in Tab. 6a to 6c.

---

> ### Author Rebuttal · Authors · 2025-07-31
>
> Dear reviewer FW6y:
>
> On behalf of all authors, we extend our sincerest gratitude to you! We deeply appreciate the valuable time and effort you dedicated to the review process, especially during this period of significantly increased submission volume. We particularly appreciate your recognition of **the breadth of our experiments and the effectiveness of our methodology**. Below, we address your questions point by point:
>
> >**Q1.** The brackets in Eq. 3 are not closed, making it unclear.
>
> **R1.** We sincerely apologize for an unfortunate error that slipped into Equation 3 during manuscript preparation—namely, the inclusion of an extraneous left parenthesis. We regret any confusion this may have caused. **The correct expression should read
> $\mathcal{L}\_{AFFS}\^{i,k}=\frac{1}{N}\sum\_ {n=1}^{N}\Vert \mathcal{F}(f\_{i}^{k}(R\_n)-f\_{i}^{k}(F\_n))-M_n \Vert \_{2}\^{2}$, where $\mathcal{F}$ denotes feature normalization and spatial interpolation to align $f_{i}^{k}$ with mask $M_n$. We will correct this in the revised manuscript.**
>
> ---
>
> >**Q2.** FIA-USA’s effectiveness likely depends heavily on the quality of its pseudo-fake generation. This creates two key vulnerabilities: 1. Performance may degrade against adversarial attacks (e.g., perturbations designed to fool the detector). 2. Performance may degrade when faced with higher-fidelity forgeries potentially emerging in the future.
>
> **R2.** We acknowledge the validity of your concerns. We  address them in two parts:
> * First, regarding adversarial attacks: Fundamentally, deepfake detection operates as a downstream classification task (e.g., image/video classification), and classification models remain highly vulnerable to adversarial attacks—including data poisoning, evasion attacks, and backdoor attacks. While many researchers are actively developing defense mechanisms against these threats, **our core contribution is orthogonal to adversarial defense. Our primary objective focuses on enhancing the model’s inherent detection performance.** That said, we find your proposed idea—developing data augmentation methods that simultaneously resist adversarial attacks and generalize to unseen counterfeits—highly valuable. This concept has been incorporated into our subsequent research plan.
> * Second, concerning the evolution of forgery methods: We fully recognize this challenge, **as the relationship between forgery and detection is inherently adversarial—an ongoing pursuit where each advance prompts countermeasures.** While our current model demonstrates strong detection capabilities, its efficacy against future generative models may diminish (as seen even with SOTA methods like SBI in detection generative methods). **Rest assured, we are committed to advancing more robust detection frameworks to address the emergent trust crisis in digital media.**
>
>
> ---
> >**Q3.** The paper lacks a formal theoretical justification for the strong cross-domain generalization performance of FIA and USA. Providing such analysis would strengthen the claims.
>
> **R3.** Thank you for your valuable feedback regarding the theoretical underpinnings of our paper. **In this work, we derive the effectiveness of our method directly from analyzing the operational processes and inherent defects of deepfake generation techniques.** Specifically, in Section 3 of the main text, we primarily discuss the **irretrievability of FIA**.  This argument stems from two fundamental constraints:
> * a. The generation bottleneck: All deepfake methods (whether full face synthesis or face swapping) require generating high-frequency facial details. However, generators are inherently limited in their ability to perfectly generate faces.
> * b. The inevitability of blend: Face swapping techniques (involving both macro-editing of 4/81 key points and micro-editing) crucially depend on blend operations(see Appendix B.1). This process forcibly concatenates heterogeneous pixels, creating local  discontinuities (like color or brightness jumps) at the boundaries, which form consistent artifact patterns across methods.
>
> Additionally, **USA are unavoidable**: whether using autoencoders (AE) or super-resolution models (SR), the generator’s decoder must perform upsampling operations. Appendix B provides a detailed taxonomy of deepfake methods, explaining the basic generation process and the specific types of artifacts each method is prone to introducing. Building on this analysis of generation defects, our method introduced the FIA-USA data augmentation strategy. This augmentation is specifically designed to simulate the various manifestations of FIA and USA, enhancing the model's ability to detect them. For instance:
> * AE and SR augmentation targets USA restoration.
> * Blending augmentation targets FIA restoration.
>
> Therefore, the justification for our method’s effectiveness is straightforward and intuitive: we start from the inherent defects of the generation process itself and use targeted data augmentation to train the model to recognize these specific artifacts more accurately. **Appendix Table 10 provides empirical support for this intuition, demonstrating that our FIA-USA augmentation covers nearly the entire spectrum of observable deepfake artifacts. In addition, Section B of the appendix provides a detailed analysis of the current mainstream forgery techniques, including the classification of forgery techniques and why they are classified in this way, the basic process of each forgery, and which artifacts are introduced at which step.**
>
>
> ---
> > **Q4.**  A detailed analysis about the results in Tab. 6a to 6c.
>
> **R4.** This section provides a detailed analysis of Table 6 (to be updated in the camera-ready version), focusing primarily on the ablation studies designed to evaluate our proposed method.
>
> * **Table 6a primarily investigates the effectiveness of each core component.** The combination of FIA-USA, AFFS, and RCR yields optimal performance. As intended in our design, AFFS and RCR are crucial for fully leveraging the potential of FIA-USA. Removing any component degrades performance, with the absence of FIA-USA having the most significant impact (a 4.15% decrease). It is important to clarify that "w/o FIA-USA" specifically refers to using only the data augmentation proposed by SBI. Furthermore, RCR exerts a stronger influence on FIA-USA's effectiveness compared to AFFS.
>
>      **The ablation specifically targeting FIA-USA (components within it) reveals:** Various data augmentation techniques contribute differently. SR and AE are shown to be effective for detecting generative forgeries. Conversely, MTMS appears to suppress performance on this specific forgery type. This aligns with our theoretical analysis: generative forgeries constitute global manipulations that produce fewer  Face Inconsistency Artifacts (FIA) but primarily exhibit  Up-Sampling Artifacts (USA). As seen from MTMS’s performance on traditional forgeries, increasing the diversity of FIA proves beneficial for detecting face swapping forgery.
>
>     **Regarding the RCR ablation:**  The results align well with our chosen hyperparameters. They indicate that the PCR contributes substantially more to the model's performance than the ICR. This observation is consistent with the hyperparameter sensitivity analysis presented in Table 6c: increasing the loss weight for PCR while decreasing that for ICR leads to significant performance gains. Conversely, increasing the ICR weight while reducing the PCR weight results in only marginal improvements.
>
> *  **Table 6b demonstrates the compatibility of our method with various network frameworks**, while also illustrating the critical impact of model parameter size and architectural design on detector performance. This highlights the ongoing importance of developing more robust and specialized deep learning architectures for counterfeit detection tasks.
> *  **Table 6c provides a detailed ablation experiment for our hyperparameter selection**, which is consistent with the ablation experiment results for RCR components in Table 6a. We find that increasing the weight of PCR loss for BCE loss significantly improves model performance, while reducing ICR for BCE loss also improves model loss. However, conversely, model performance will not be significantly improved.
>
>
> ---
> > **Q5.** Why was the current hyperparameter setting selected？
>
> **R5.**  **We conducted detailed experiments to determine the values of our hyperparameters.**  As discussed in the fourth point, our findings demonstrate that increasing the weight of PCR while decreasing the weight of ICR relative to the BCE loss enhances model performance. Conversely, applying the opposite weighting scheme reduces performance. Guided by these results, we determine the final hyperparameter settings after extensive experimentation. **Table 6.c displays  our experimental results.**
>
> We attribute this outcome to a key observation: ICR images are subjected to different data augmentations (e.g., compression, color shifts, contrast adjustments), resulting in significantly reduced feature-level similarity relative to PCR. Consequently, ICR is less effective than PCR, justifying its lower weighting.

---

> > ### Comment · Reviewer_FW6y · 2025-08-04
> >
> > I appreciate the clear rebuttal made by the authors. After considering the authors' response, I decided to maintain my original score.

---

> > > ### Author Response · Authors · 2025-08-05
> > >
> > > We appreciate your recognition of our work and your dedication. Your feedback greatly benefits us, and we are pleased to have addressed the issues you raised in our paper.

---

### Official Review · Reviewer_fmVY · 2025-07-01

**Clarity:** 3
**Significance:** 3
**Originality:** 3
**Rating:** 5
**Confidence:** 4

**Summary:**

While this paper contains minor limitations, they do not detract from its overall strengths. The logical flow is exceptionally clear, experimental validation is comprehensive with impressive performance, and supplementary materials (particularly the visual analysis of misclassified samples and systematic evaluation of mainstream deepfake methods) provide substantial value. I therefore recommend a Weak Accept. Should the authors address Concern #1 and #2 raised in my review, I would be more willing to elevate my score.

**Questions:**

1.  It is recommended to relocate selected content from the appendix to the main body.
2.  It is necessary to clarify the details of video-level detection.

**Ethical Concerns:**

["NO or VERY MINOR ethics concerns only"]

**Final Justification:**

I maintain my original score that recommend the paper to be accepted.

**Quality:**

3

**Strengths And Weaknesses:**

Strenth:
This paper categorizes deepfake artifacts into two types: FIA (Face Inconsistency Artifacts): Arising from detail discrepancies between forged faces and backgrounds (e.g., lighting/texture discontinuities), USA (Up-Sampling Artifacts): Introduced by the up-sampling process in generator decoders (e.g., interpolation traces). By analyzing technical workflows of mainstream forgery algorithms (e.g., Full Face Synthesis and Face Swapping), it demonstrates that all existing methods inevitably produce at least one of FIA/USA artifacts. Building on this foundation, the paper proposes FIA-USA data augmentation, whose three components (Multi-Type Mask Synthesis (MTMS), Multi-Modal Reconstruction (MMR), Random Artifact Combination (RAC)), collectively cover potential artifact types in deepfakes comprehensively, addressing the limitation of prior augmentation methods that only simulate FIA locally or globally. Additionally, the complementary AFFS and RCR techniques fully leverage FIA-USA’s capabilities. Validation across 7 datasets (including traditional and generative deepfakes) encompassing 58 distinct forgery methods proves the approach’s generalizability. Though the framework involves multiple components, the authors’ thorough ablation studies confirm each element’s effectiveness, and the paper’s logical structure remains exceptionally clear.

Weakness:
1.The FIA-USA framework involves numerous components, and understanding its full operational procedure necessitates referring to the algorithmic details in the appendix. To improve accessibility, the complete algorithm should be moved to the main text. Additionally, the initialization process of AFFS relies on training dataset data—a critical implementation detail currently buried in the appendix that warrants inclusion in the main body.

2.Although the proposed method still relies on the frameworks of SBI and Face X-ray, the improvements derived from the authors' artifact analysis of mainstream deepfake techniques are compelling. As clearly demonstrated in Table 1, their approach exhibits significant distinctions from prior works. However, I am perplexed: the method appears to be an image-domain enhancement—why can it be applied to video-level deepfake detection? More specifically, I wonder whether the video-level experiments use the maximum value or the average value of multi-frame results.


3.While the method demonstrates compelling efficacy across most forgery techniques, its detection performance remains notably weak for certain generative approaches, particularly StyleGANXL, DIT, and others. Overall, the performance is still impressive.

4.The framework contains too many components, making the method somewhat complex. Although the author has devoted a lot of effort to elaborating on each component, it still needs to be carefully read in conjunction with the appendix and have a certain understanding of data augmentation work in the RGB field.

---

> ### Author Rebuttal · Authors · 2025-07-31
>
> Dear reviewer fmVY:
>
> On behalf of all authors, we deeply appreciate your contributions during the review process. We are honored to benefit from your feedback and appreciate your recognition of our **article's logical framework, experimental richness, and model performance**. Below, we address your questions point by point:
>
> >**Q1.** Move the full FIA-USA algorithm and the initialization details of AFFS from the appendix to the main text for better accessibility and emphasis, as they are critical to understanding the framework.
> >
> **R1.** Regarding the manuscript's presentation, we acknowledge your observation about the inherent complexity of our methodology and the extensive experimental results.
> Our proposed universal deepfake detection framework comprises three  core components, each vital for robust detection performance:
> * The FIA-USA dual-artifact collaborative enhancement framework, which generates pseudo-fake samples containing Face Inconsistency Artifacts (FIA) and Up-Sampling Artifacts (USA) through Multi-Type Mask Synthesis (MTMS) for hierarchical mask generation (covering macro-editing boundaries via convex hulls of 4/81 keypoints and micro-editing traces via eye/mouth feature combinations), Multi-Modal Reconstruction (MMR) using autoencoders and super-resolution models to simulate USA by reconstructing target faces, and Random Artifact Combination (RAC) to blend source and reconstructed images with randomly selected masks and artifacts;
> * Automatic Forgery-aware Feature Selection (AFFS), which reduces feature redundancy in Feature Pyramid Networks by ranking channels based on normalized response discrepancies between real and forged regions to retain artifact-sensitive dimensions;
> * Region-aware Contrastive Regularization (RCR), which enhances discrimination between manipulated and authentic areas through patch-level contrast (minimizing feature distances in real faces while maximizing them in fakes) and image-level contrast (aligning real regions across samples).
>
> **To enable the clearest possible description of each component given the inherent complexity and to adhere to the strict page limits for the main text while ensuring its overall quality**, some in-depth analyses, supplementary algorithmic details, and extensive ablation studies are necessarily included in the appendix in this submission version. **Should our paper be accepted, we will relocate the most critical of this appendix content back into the main text in the camera-ready version to provide the most comprehensive presentation possible within the final page constraints**.
>
> ---
>
> >**Q2.** The proposed method's compelling improvements—derived from the authors' artifact analysis of mainstream deepfake techniques—are notable, even though it relies on the SBI and Face X-ray frameworks. Described as image-domain enhancement, a key question remains: how is this approach applicable to video-level detection?
>
> **R2.** While traditional blend-based augmentation methods like Face X-ray and SBI primarily focus on simulating isolated Face Inconsistency Artifacts (FIA) through either global or local facial manipulations—thereby limiting their ability to capture the complex artifact interplay in real deepfakes. Our FIA-USA framework pioneers a dual-artifact collaborative strategy that simultaneously addresses both FIA and Up-Sampling Artifacts (USA), which stem from inherent decoder operations in all deepfake generators.
>
> Specifically, we overcome previous limitations through three key innovations:
> 1) Multi-Type Mask Synthesis that  models both macro-editing boundaries (via 4/81-keypoint convex hulls) and micro-editing traces (through randomized eye/mouth feature combinations).
> 2) Multi-Modal Reconstruction that employs autoencoder and super-resolution model to systematically simulate USA patterns across diverse generator architectures.
> 3) Random Artifact Combination that blends reconstructed variants with source images to create comprehensive pseudo-fakes covering global-local and facial-regional inconsistencies. In Table 1, we have conducted a detailed comparison with previous data augmentation methods.
>
> For video-level analysis, our methodology employs uniform sampling of 32 frames per video. We compute frame-wise forgery probabilities independently through our detection pipeline, then derive the final video-level confidence score by **calculating the mean of all sampled frame probabilities.**
>
> **Table 1: Comparison of our framework and state-of-the-art(SOTA) methods using pseudo-deepfake synthesis. Our method differs from previous methods in terms of the level at which alterations are applied (local vs global), the introduced artifacts, and the level at which facial inconsistencies are applied.**
>
> | | global  | local | face-level |region-level | Up-Sampling |
> | :--- | :--: | :--: | :--: | :--: | :---: |
> |**Face Xray, PCL, OST**   |$\checkmark$  |  | $\checkmark$  | |  |
> | **SLADD** | $\checkmark$  |  | | $\checkmark$  |  |
> | **SBI** | $\checkmark$  |  | $\checkmark$ |  | |
> | **SeeABLE** | | $\checkmark$  |  | $\checkmark$  |  |
> | **RBI** | $\checkmark$ | | $\checkmark$  |  | |
> | **Plug-and-Play** | | $\checkmark$  | |$\checkmark$   |  |
> | **Ours** | $\checkmark$  |$\checkmark$ | $\checkmark$  | $\checkmark$  | $\checkmark$  |
>
> ---
>
> >**Q3.**  While the method demonstrates compelling efficacy across most forgery techniques, its detection performance remains notably weak for certain generative approaches, particularly StyleGANXL, DIT, and others. Overall, the performance is still impressive.
>
> **R3.**  We sincerely appreciate your recognition of the overall efficacy of our method across diverse forgery techniques. As highlighted in our comprehensive evaluation (Section 4), we rigorously evaluate our deepfake detection framework across seven benchmark datasets, collectively spanning over 58 distinct forgery methods to ensure comprehensive validation of generalization capabilities. These include five traditional deepfake datasets—FaceForensics++ (FF++), Deepfake Detection (DFD), Deepfake Detection Challenge (DFDC), DFDC Preview (DFDCP), and CelebDF (CDF)—which cover classical manipulation techniques like identity swapping and expression reenactment. Additionally, two generative deepfake datasets—Diffusion Facial Forgery (DiFF) and DF40 are used to test robustness against modern threats, with DiFF incorporating 13 text/image-driven synthesis methods (e.g., Stable Diffusion, LoRA) and DF40 encompassing 40 cutting-edge techniques spanning face swapping, full-face synthesis, and fine-grained attribute editing. While our method demonstrates robust detection performance for 53 of the 58 techniques evaluated（As shown in Table 9）, we acknowledge the relatively weaker efficacy for some generative approaches such as StyleGANXL and DiT. **We plan to further improve our work in the future and develop more robust detection methods.**
>
> ---
> >**Q4.** The framework contains too many components, making the method somewhat complex. Although the author has devoted a lot of effort to elaborating on each component, it still needs to be carefully read in conjunction with the appendix and have a certain understanding of data augmentation work in the RGB field.
>
> **R4.** Thank you for recognizing our paper. Our work does have numerous components, but it is not simply a pile up. Each component plays an irreplaceable role in our methodology.
>
> Our motivation for starting this paper is to notice a phenomenon: The fundamental challenge driving deepfake detection development is the inherent limitation of existing deepfake detectors in generalizing across diverse generation techniques (e.g., GANs, Diffusion models). These methods produce highly variable artifacts, causing specialized detectors to fail when encountering unseen manipulation methods. Through systematic analysis of mainstream forgery algorithms, we identity two universal flaws underlying all deepfakes: 1) Face Inconsistency Artifacts (FIA), stemming from generators’ inability to perfectly harmonize facial details with background physical properties (e.g., lighting/texture), exacerbated by blending boundaries in face-swapping techniques; and 2) Up-Sampling Artifacts (USA), mathematical distortions introduced during decoder-based image reconstruction in encoder-decoder architectures.
>
>  Then，We propose FIA-USA with the aim of developing more robust forgery detection models by simulating these two more common features through data augmentation. At the same time, we notice that work such as SBI focuses more on the data augmentation process itself, which  makes us doubt  whether FIA-USA's capabilities are fully utilized?
>
> Therefore, we propose RCR (Region aware Contrastive Regularization). It amplifies three critical distinctions: 1) intra-real-sample consistency, 2) intra-fake-sample discrepancies between manipulated/authentic regions, and 3) cross-sample consistency in real regions. This forces the model to pinpoint USA distribution  and FIA boundary discontinuities, enhancing sensitivity to artifacts.
>
> Complementing this, AFFS (Automatic Forgery-aware Feature Selection) addresses feature redundancy in standard Feature Pyramid Networks (FPN). It purifies the feature space by using forgery-region masks as supervision signals to quantify each channel’s sensitivity to USA/FIA, selectively retaining artifact-correlated dimensions.
>
> In addition, consistent with your point of view, we  recommend readers to read a series of previous works on data augmentation, including Face X-ray, SBI, RBI, etc., because these works are progressive layer by layer, and reading previous works can indeed help better understand our method.

---

> > ### Comment · Reviewer_fmVY · 2025-08-05
> >
> > Thanks to the authors for the thoughtful response.
> > I am glad to see that most of my concerns raised in the previous review have been addressed.
> > I have no further comments and maintain my score.

---

> > > ### Author Response · Authors · 2025-08-05
> > >
> > > We appreciate your recognition of our work and are pleased to address your concerns about our paper. The questions you raised are valuable for improving our work, and we thank you for your time and efforts during the review period.

---

### Official Review · Reviewer_cBZ3 · 2025-07-01

**Clarity:** 4
**Significance:** 3
**Originality:** 3
**Rating:** 5
**Confidence:** 3

**Summary:**

This paper tackles the growing problem of deepfakes by suggesting a more general way to detect them, rather than trying to identify every specific forgery technique. The authors noticed that most deepfakes, regardless of how they’re created, tend to have two main types of flaws: Face Inconsistency Artifacts (FIA) and Up-Sampling Artifacts (USA).

The author generates "pseudo-fake" training images that deliberately include only these two types of artifacts. The idea is that by training a detector on these fundamental flaws, it becomes much better at spotting real deepfakes it hasn't seen before. They also added a couple of supporting techniques: Automatic Forgery-aware Feature Selection (AFFS) helps the detector zero in on the most relevant forgery clues, and Region-aware Contrastive Regularization (RCR) helps it distinguish between genuine and manipulated areas. Essentially, they're showing that a focused approach on universal artifacts can lead to a more robust and adaptable deepfake detection system.

**Questions:**

In the limitations, the authors acknowledge "relatively weaker performance in specific edge cases" among the 58+ forgery methods tested. Could they provide more specific examples of which forgery methods constitute these "edge cases"?

**Ethical Concerns:**

["NO or VERY MINOR ethics concerns only"]

**Final Justification:**

The authors provided detailed answers to my questions, and I have decided to increase the paper's score.

**Limitations:**

Yes

**Quality:**

3

**Strengths And Weaknesses:**

Strengths:
1) This paper seems pretty solid in terms of quality. The authors put in the work to systematically analyze existing deepfake algorithms to pinpoint common artifacts, which is a smart move. They propose a novel pseudo-fake creation framework that's designed specifically to introduce these general artifacts, avoiding less common ones, which sounds really intentional and well-thought-out.
2) The paper definitely brings some novel ideas to the table. While data augmentation for deepfake detection isn't new, their specific approach of categorizing artifacts into FIA and USA and then explicitly designing a pseudo-fake generation framework (FIA-USA) to introduce both types is novel. They highlight that existing methods typically focus on isolated artifact types or single-scale modifications, which limits their generalization. Their Multi-Type Mask Synthesis for blending and Multi-Modal Reconstruction for USA, leveraging autoencoders and super-resolution, seem quite original in their combination and purpose.

Weaknesses:
1) While the FIA-USA framework and its components are innovative, the underlying concepts of blending and reconstruction for data augmentation have been explored before in deepfake detection.
2) While the experiments are extensive, they mention that some results are "cited from" or "our reproduction results" for other methods.

---

> ### Author Rebuttal · Authors · 2025-07-31
>
> Dear Reviewer cBZ3:
>
> On behalf of all authors, we sincerely appreciate the significant time and effort you dedicated to reviewing our manuscript.  Your thorough, thoughtful, and insightful feedback is invaluable to our work, and we are deeply grateful for your recognition of our **rigorous experiments and innovative methodology**. Below, we address your questions point by point:
>
> > **Q1.** While the FIA-USA framework and its components are innovative, the underlying concepts of blending and reconstruction for data augmentation have been explored before in deepfake detection.
>
> **R1.** First, regarding the innovation of our data augmentation approach: While data augmentation has long been proposed in forgery detection research and **remains a vibrant research frontier**, existing methods predominantly focus on isolated artifact types and single blend regions.
>
> **As comprehensively detailed in Table 1 of the paper**, our method provides substantially broader coverage across both artifact diversity and blend region complexity. This methodological advancement is rigorously grounded in our systematic analysis of prevalent forgery techniques (Appendix B), where we categorize forgery artifacts into Face Inconsistency Artifacts (FIA) and  Up-Sampling Artifacts (USA).
>
> This taxonomy revitalizes data augmentation strategies and extends their applicability to previously undetectable forgery methods—particularly generative forgeries. Extensive experimental validation confirms our approach's efficacy.
>
> **Table 1: Comparison of our framework and state-of-the-art(SOTA) methods using pseudo-deepfake synthesis. Our method differs from previous methods in terms of the level at which alterations are applied (local vs global), the introduced artifacts, and the level at which facial inconsistencies are applied.**
>
>
> | | global  | local | face-level |region-level | Up-Sampling |
> | :--- | :--: | :--: | :--: | :--: | :---: |
> |**Face Xray, PCL, OST**   |$\checkmark$  |  | $\checkmark$  | |  |
> | **SLADD** | $\checkmark$  |  | | $\checkmark$  |  |
> | **SBI** | $\checkmark$  |  | $\checkmark$ |  | |
> | **SeeABLE** | | $\checkmark$  |  | $\checkmark$  |  |
> | **RBI** | $\checkmark$ | | $\checkmark$  |  | |
> | **Plug-and-Play** | | $\checkmark$  | |$\checkmark$   |  |
> | **Ours** | $\checkmark$  |$\checkmark$ | $\checkmark$  | $\checkmark$  | $\checkmark$  |
>
> ---
>
> > **Q2.** While the experiments are extensive, they mention that some results are "cited from" or "our reproduction results" for other methods.
>
> **R2.**  Please allow me to reiterate our gratitude for your suggestions regarding the experimental content. We sincerely appreciate your meticulous review of our paper. All experiments in our study primarily focus on evaluating our method's robustness against various forgery techniques, while simultaneously comparing it with state-of-the-art forgery detection methods to demonstrate its superiority.
>
> However, a critical issue arises: although different studies use the same datasets, their experimental settings often differ. For instance, SBI employs 8 frames for training, whereas methods like LAA-Net utilize 128 frames. Such discrepancies in experimental configurations lead to unfair comparisons. **In this work, reproducing all existing methods' experiments is not our objective. Our goal is to ensure maximally equitable experimental conditions.** Many prior studies, such as DeepfakeBench, have addressed this challenge.
>
> To minimize workload and focus on the paper's overall presentation, we  extensively reuse results obtained under identical experimental settings. Additionally, for papers not previously reproduced with consistent settings, we conduct independent re-implementations. Consequently, many experimental results are cited from other publications. **When incorporating such data, we rigorously verify the alignment of experimental settings.**
>
>
> ---
>
>
> > **Q3.** The paper acknowledges 'relatively weaker performance in specific edge cases' across 58+ forgery methods. Please explicitly state which specific forgery methods constitute these edge cases.
> >
> **R3.** Our approach demonstrates robust performance against 53 out of 58 evaluated forgery techniques while showing limitations on only 5 specific methods.  **As noted in Appendix Table 9**, we highlight successful results through **bolding and underlining**, which may inadvertently obscure visibility of underperforming cases.
>
> These less effective scenarios primarily involve: **Which Face Is Real, StarGANv2, StyleGAN-XL,HeyGen, DiT**. Although our method shows reduced efficacy against these particular forgery techniques, its strong performance against the majority of tested methods remains encouraging. We will prioritize optimizing performance for these specific cases in future work.
>
> **Table 9:**  **We test all forgery techniques provided on the DF40 dataset and reported Frame-Level AUC.** All results with AUC greater than 80 are **highlighted** in bold, while those with AUC less than 80 but greater than 70 are **underlined**.
>
>  **Type** | mobileswap    | MidJourney    | faceswap        | styleclip        | DiT             | lia               | ddim             | mcnet            | RDDM             | StyleGAN3
> :--------:|:-------------:|:-------------:|:---------------:|:----------------:|:---------------:|:-----------------:|:----------------:|:----------------:|:----------------:|:-----------------------------:
>  **AUC**  | $\textbf{0.97}$ | $\textbf{0.97}$ |$\textbf{0.89}$   | $\textbf{0.82}$   | 0.66            | $\textbf{0.91}$     | $\textbf{0.97}$    | $\textbf{0.84}$    | $\underline{0.70}$ | $\textbf{0.93}$
>  **Type** | e4e           | pixart        | whichfaceisreal | deepfacelab      | StyleGANXL      | heygen            | facedancer       | MRAA             | pirender         | VQGAN
>  **AUC**  | $\textbf{0.95}$ | $\textbf{0.93}$ | 0.42            | $\underline{0.77}$ | 0.41            | 0.58              | $\textbf{0.83}$    | $\textbf{0.85}$    | $\textbf{0.81}$    | $\textbf{0.85}$
>  **Type** | StyleGAN2     | facevid2vid   | simswap         | inswap           | one\_shot\_free | wav2lip           | e4s              | starganv2        | tpsm             | sd2.1
>  **AUC**  | $\textbf{0.93}$ | $\textbf{0.83}$ | $\textbf{0.91}$   | $\textbf{0.87}$    | $\textbf{0.85}$   | $\underline{0.76 }$ | $\textbf{0.88}$    | 0.48             | $\textbf{0.83}$    | $\textbf{0.97}$
>  **Type** | blendface     | hyperreenact  | uniface         | CollabDiff       | fsgan           | danet             | SiT              | sadtalker        | fomm             |
>  **AUC**  | $\textbf{0.94}$ | $\textbf{0.80}$ | $\textbf{0.92}$   | $\textbf{0.95}$    | $\textbf{0.86}$   | $\textbf{0.80}$     | $\underline{0.72}$ | $\underline{0.74}$ | $\textbf{0.85}$    |

---

### Comment · Area_Chair_HggL · 2025-08-04
**Discussion**

Dear Reviewers,

Thank you very much for your time and efforts. As we are approaching the deadline, we kindly ask you to review the rebuttal and share any remaining concerns with the authors for discussion.

Best regards,

AC

---

### Note · Authors · 2025-08-12

Dear Area Chair,

We sincerely thank all reviewers for their constructive comments and recognition of our work.
* **[Motivation & Novelty]** (Reviewers fmVY, FW6y, uxGL,cBZ3 )
* **[Extensive Validation]** (Reviewers  fmVY, FW6y, uxGL,cBZ3 )
* **[Detailed ablation experiments]** (Reviewers  fmVY, FW6y, uxGL)
* **[Robustness]** (Reviewers FW6y, uxGL)


Our work, **FIA-USA**, proposes a novel image-level framework for universal deepfake detection. By constructing pseudo-fake samples containing Face Inconsistency Artifacts (FIA) and Up-Sampling Artifacts (USA), our method achieves robust generalization to unseen techniques.

Through the discussion period, we have addressed all the major concerns raised by reviewers (Reviewers **fmVY, FW6y, uxGL** increased/maintained to accept). We also hope we have sufficiently addressed Reviewer  **cBZ3**'s concern, as no futher concern were raised after our response.

We hope this work help address some challenges in current detection methods and contributes to the broader research community.

Thank you once again for your time and careful consideration.

Sincerely,

The Authors of Submission 19908

---

### Decision · Program_Chairs · 2025-09-17

**Decision:**

Accept (poster)

**Comment:**

This paper addresses the urgent challenge of generalizable deepfake detection. Rather than designing detectors for every forgery technique, the authors identify two universal artifact categories, Face Inconsistency Artifacts (FIA) and Up-Sampling Artifacts (USA), that inevitably arise across diverse deepfake generators. Building on this insight, they propose FIA-USA, a pseudo-fake creation framework that systematically synthesizes training data containing only these artifacts, and complement it with Automatic Forgery-aware Feature Selection (AFFS) and Region-aware Contrastive Regularization (RCR).

The work is novel and well-motivated, providing a principled taxonomy of artifacts and a carefully designed augmentation pipeline. Experiments are comprehensive and rigorous, spanning seven benchmark datasets (traditional and generative, with over 58 forgery methods), and show consistent improvements in generalization and robustness. Ablation studies and visual analyses convincingly demonstrate the contributions of each component.

While the framework is complex and exhibits somewhat weaker performance on a few modern generative forgery methods (e.g., StyleGANXL, DiT), these are acknowledged by the authors and do not diminish the overall impact. Reviewers found the rebuttal clarifications satisfactory, particularly regarding novelty and video-level detection methodology.

Overall, this is a technically solid and impactful paper. It introduces a principled perspective on universal artifacts, demonstrates state-of-the-art performance, and provides extensive validation. I recommend acceptance.